# NuRD and CAF-1-mediated silencing of the D4Z4 array is modulated by DUX4-induced MBD3L proteins

Amy E Campbell[1]*, Sean C Shadle[1,2], Sujatha Jagannathan[1,3,4†], Jong-Won Lim[1], Rebecca Resnick[1,2,5], Rabi Tawil[6], Silvère M van der Maarel[7], Stephen J Tapscott[1,8]*

[1]Human Biology Division, Fred Hutchinson Cancer Research Center, Seattle, United States; [2]Molecular and Cellular Biology Program, University of Washington, Seattle, United States; [3]Basic Sciences Division, Fred Hutchinson Cancer Research Center, Seattle, United States; [4]Computational Biology Program, Public Health Sciences Division, Fred Hutchinson Cancer Research Center, Seattle, United States; [5]Medical Scientist Training Program, University of Washington, Seattle, United States; [6]Department of Neurology, University of Rochester Medical Center, Rochester, United States; [7]Department of Human Genetics, Leiden University Medical Center, Leiden, Netherlands; [8]Department of Neurology, University of Washington, Seattle, United States

**\*For correspondence:**
acampbel@fredhutch.org (AEC);
stapscot@fredhutch.org (SJT)

**Present address:** †Department of Biochemistry and Molecular Genetics, RNA Bioscience Initiative, University of Colorado School of Medicine, Aurora, United States

**Competing interests:** The authors declare that no competing interests exist.

**Abstract** The DUX4 transcription factor is encoded by a retrogene embedded in each unit of the D4Z4 macrosatellite repeat. DUX4 is normally expressed in the cleavage-stage embryo, whereas chromatin repression prevents DUX4 expression in most somatic tissues. Failure of this repression causes facioscapulohumeral muscular dystrophy (FSHD) due to mis-expression of DUX4 in skeletal muscle. In this study, we used CRISPR/Cas9 engineered chromatin immunoprecipitation (enChIP) locus-specific proteomics to characterize D4Z4-associated proteins. These and other approaches identified the Nucleosome Remodeling Deacetylase (NuRD) and Chromatin Assembly Factor 1 (CAF-1) complexes as necessary for DUX4 repression in human skeletal muscle cells and induced pluripotent stem (iPS) cells. Furthermore, DUX4-induced expression of MBD3L proteins partly relieved this repression in FSHD muscle cells. Together, these findings identify NuRD and CAF-1 as mediators of DUX4 chromatin repression and suggest a mechanism for the amplification of DUX4 expression in FSHD muscle cells.
DOI: https://doi.org/10.7554/eLife.31023.001

## Introduction

Repetitive DNA sequences make up the majority of the human genome (*Birney et al., 2007*; *de Koning et al., 2011*), and these ubiquitous but understudied elements play a critical role in important biological processes such as embryogenesis and cellular reprogramming (*Chuong et al., 2017*; *Elbarbary et al., 2016*; *Gerdes et al., 2016*). For example, each unit of the D4Z4 macrosatellite repeat array contains a copy of the double homeobox 4 (*DUX4*) retrogene that is expressed in the germline and in four-cell human embryos where DUX4 activates a cleavage-specific transcriptional program (*De Iaco et al., 2017*; *Hendrickson et al., 2017*; *Snider et al., 2010*; *Whiddon et al., 2017*). This is in contrast to somatic tissues where DUX4 is silenced via repeat-mediated epigenetic repression of the D4Z4 arrays (*Das and Chadwick, 2016*; *Daxinger et al., 2015*; *Snider et al., 2010*; *van Overveld et al., 2003*; *Zeng et al., 2009*). To date, little is understood

**eLife digest** The DNA sequences of humans and other mammals contain many repetitive regions. This repetition makes these regions difficult to study with conventional approaches, and so the exact role of repetitive DNA is not fully understood.

A particular sequence of repetitive DNA that plays an important role in human health contains a gene called DUX4 in each repeat. DUX4 is normally active in stem cells and in early-stage embryos. This gene is then switched off or 'silenced' during later stages of development and in most cells of the body. However, in some individuals the DUX4 gene inappropriately activates in muscle cells. This causes a disease known as facioscapulohumeral muscular dystrophy (FSHD), in which muscle weakness begins in the face and upper body and eventually spreads to other muscles. Currently, there is no cure for FSHD.

Proteins that bind to DNA can control the activity of nearby genes. Little is known about which proteins silence DUX4 at the appropriate time and in the right cells, so Campbell et al. set out to identify the proteins that attach to the repetitive DNA sequences containing DUX4. Further investigation showed that several of these proteins play an important role in keeping DUX4 turned off, including two protein complexes called NuRD and CAF-1. These complexes are necessary to silence DUX4 in human muscle cells and stem cells. Campbell et al. also identified a protein that can increase the activity of the DUX4 gene in FSHD muscle cells by overcoming the silencing activity of the NuRD complex.

Overall, the results presented by Campbell et al. provide the groundwork for developing new treatments for FSHD. The next step will be to discover ways of enhancing the ability of NuRD and CAF-1 to silence the DUX4 gene.

DOI: https://doi.org/10.7554/eLife.31023.002

about how the epigenetic repression of DUX4 is relieved at specific times during germline and early embryo development, or what the mechanisms of establishing and maintaining epigenetic repression during later development and in somatic tissues are.

Facioscapulohumeral muscular dystrophy (FSHD) is caused by the mis-expression of DUX4 in skeletal muscle (*Tawil et al., 2014*) and provides an experimentally tractable context in which to identify mechanisms that normally repress DUX4 in somatic cells as well as mechanisms that might regulate this repression during development. In individuals with FSHD, the epigenetic repression of DUX4 is incomplete as a consequence of having fewer than 11 D4Z4 repeats (FSHD type 1, FSHD1) or mutations in trans-acting chromatin repressors of D4Z4 (FSHD type 2, FSHD2), either of which results in ectopic expression of DUX4 in skeletal muscle when combined with a permissive chromosome 4qA haplotype that provides a polyadenylation site for the DUX4 mRNA (*Lemmers et al., 2012*; *Lemmers et al., 2010*; *van den Boogaard et al., 2016*). The mis-expression of DUX4 in skeletal muscle has many consequences that include induction of a cleavage-stage transcriptional program, suppression of the innate immune response and nonsense-mediated RNA decay (NMD) pathways, inhibition of myogenesis, and induction of cell death through mechanisms that involve the accumulation of aberrant and double-stranded RNAs (*Bosnakovski et al., 2008*; *Feng et al., 2015*; *Geng et al., 2012*; *Kowaljow et al., 2007*; *Rickard et al., 2015*; *Shadle et al., 2017*; *Snider et al., 2009*; *Wallace et al., 2011*; *Winokur et al., 2003*; *Young et al., 2013*). These cellular insults lead to progressive muscle weakness initiating in the face and upper body but eventually involving nearly all skeletal muscle groups (*Tawil et al., 2014*).

Previous studies investigating D4Z4 repeat-mediated epigenetic repression have shown the D4Z4 arrays to be silenced through multiple mechanisms, including DNA methylation and the repressive histone modifications di/trimethylation of histone H3 at lysine 9 (H3K9me2/3) and trimethylation of histone H3 at lysine 27 (H3K27me3) along with their binding proteins CBX3/HP1γ and EZH2 (*Cabianca et al., 2012*; *Huichalaf et al., 2014*; *van den Boogaard et al., 2016*; *van Overveld et al., 2003*; *Zeng et al., 2009*). Other repressor proteins have been shown to be associated with the D4Z4 repeat, including DNMT3B, SMCHD1, cohesin, CTCF, HDAC3 and a YY1/HMGB2/NCL complex (*Gabellini et al., 2002*; *Huichalaf et al., 2014*; *Lemmers et al., 2012*; *Ottaviani et al., 2009*; *van den Boogaard et al., 2016*; *Zeng et al., 2009*). In addition, DICER/AGO-dependent

siRNA-directed silencing has also been demonstrated to play a role in repressing the D4Z4 array (*Lim et al., 2015*; *Snider et al., 2009*). The genetic lesions that cause FSHD disrupt these regulatory pathways resulting in D4Z4 DNA hypomethylation; reduced H3K9me2/3 and H3K27me3 levels; and loss of HP1γ, EZH2, SMCHD1 and cohesin binding; which together culminate in ectopic DUX4 expression (*Cabianca et al., 2012*; *Daxinger et al., 2015*; *Jones et al., 2014*; *Lemmers et al., 2012*; *van den Boogaard et al., 2016*; *van Overveld et al., 2003*; *Zeng et al., 2009*). Although each of the above-mentioned studies tested specific factors based on knowledge of their role in chromatin, to date no studies have taken an agnostic approach to identify how these individual components might be integrated into repressive complexes or to understand how these complexes might be regulated.

Here, we report a locus-specific proteomics-based characterization of proteins that bind the D4Z4 array in human myoblasts and identify the NuRD and CAF-1 complexes as individually necessary to maintain DUX4 repression in skeletal muscle and induced pluripotent stem (iPS) cells. Further, we show that DUX4-mediated induction of the MBD3L family of factors relieves this repression and amplifies DUX4 expression. Together, these findings identify multiprotein complexes that regulate DUX4 expression and reveal a process for DUX4 amplification in FSHD muscle cells that provides a new candidate target for therapeutics.

## Results

### enChIP-MS identifies NuRD complex components as D4Z4 repeat-associated proteins

To identify regulators of the D4Z4 macrosatellite repeat, we carried out engineered DNA-binding molecule-mediated chromatin immunoprecipitation (enChIP) followed by mass spectrometry (MS) (enChIP-MS) (*Fujita and Fujii, 2013*) (*Figure 1A*). We transduced human MB135 control (non-FSHD) myoblasts with a lentiviral vector co-expressing FLAG-tagged, nuclease-deficient Cas9 (FLAG-dCas9) and guide RNA (gRNA) targeting the 3' end (gD4Z4-1), middle (gD4Z4-2) or 5' end (gD4Z4-3) of the D4Z4 unit, or the MYOD1 distal regulatory region (DRR) (gMYOD1) for comparison. After confirming the expression, subcellular localization, and specific chromatin occupancy of FLAG-dCas9 in each cell line (*Figure 1—figure supplement 1*), complexes containing FLAG-dCas9 were immunoprecipitated and subjected to liquid chromatography-tandem mass spectrometry for protein identification.

A total of 261 proteins were identified (*Supplementary file 1*), including known D4Z4-associated factors SMCHD1, CBX3/HP1γ and the cohesin complex components SMC1A, SMC3, RAD21 and PDS5B (*Lemmers et al., 2012*; *Zeng et al., 2009*) (*Table 1*). BRD3 and BRD4 were also identified (*Supplementary file 1*) and BET inhibitor compounds have recently been shown to regulate D4Z4 repression (*Campbell et al., 2017*). D4Z4-bound proteins were enriched in gene ontology categories that included telomere maintenance and chromatin silencing (*Supplementary file 2*), consistent with the subtelomeric localization and transcriptionally repressed state of the D4Z4 array. Strikingly, CHD4, HDAC2, MTA2 and RBBP4, which comprise many of the components of the Nucleosome Remodeling Deacetylase (NuRD) complex (*Basta and Rauchman, 2015*), were among the isolated proteins (*Table 1*). While each of these factors was identified as associated with the D4Z4 repeat in more than one gD4Z4 sample, they were either absent or present in only a single replicate from the gMYOD1 pulldowns (*Supplementary file 1*).

Occupancy of CHD4, HDAC2 and MTA2 at the D4Z4 array was confirmed by chromatin immunoprecipitation (ChIP) in MB2401 myoblasts, an independent control muscle cell line (*Figure 1B–D*). The NuRD complex can be recruited to methylated DNA by the MBD2 subunit (*Le Guezennec et al., 2006*; *Zhang et al., 1999*), and indeed, ChIP showed MBD2 enrichment at the D4Z4 region in MB2401 control myoblasts (*Figure 1E*). Together, these data demonstrate that the D4Z4 macrosatellite repeat is bound by the MBD2/NuRD complex in control human muscle cells.

### MBD2/NuRD complex components mediate transcriptional repression of the D4Z4 array

The NuRD complex represses gene transcription via the concerted effort of the core subunits HDAC1 and HDAC2; CHD3 or CHD4; MBD2 or MBD3; MTA1, MTA2 or MTA3; RBBP4 and RBBP7;

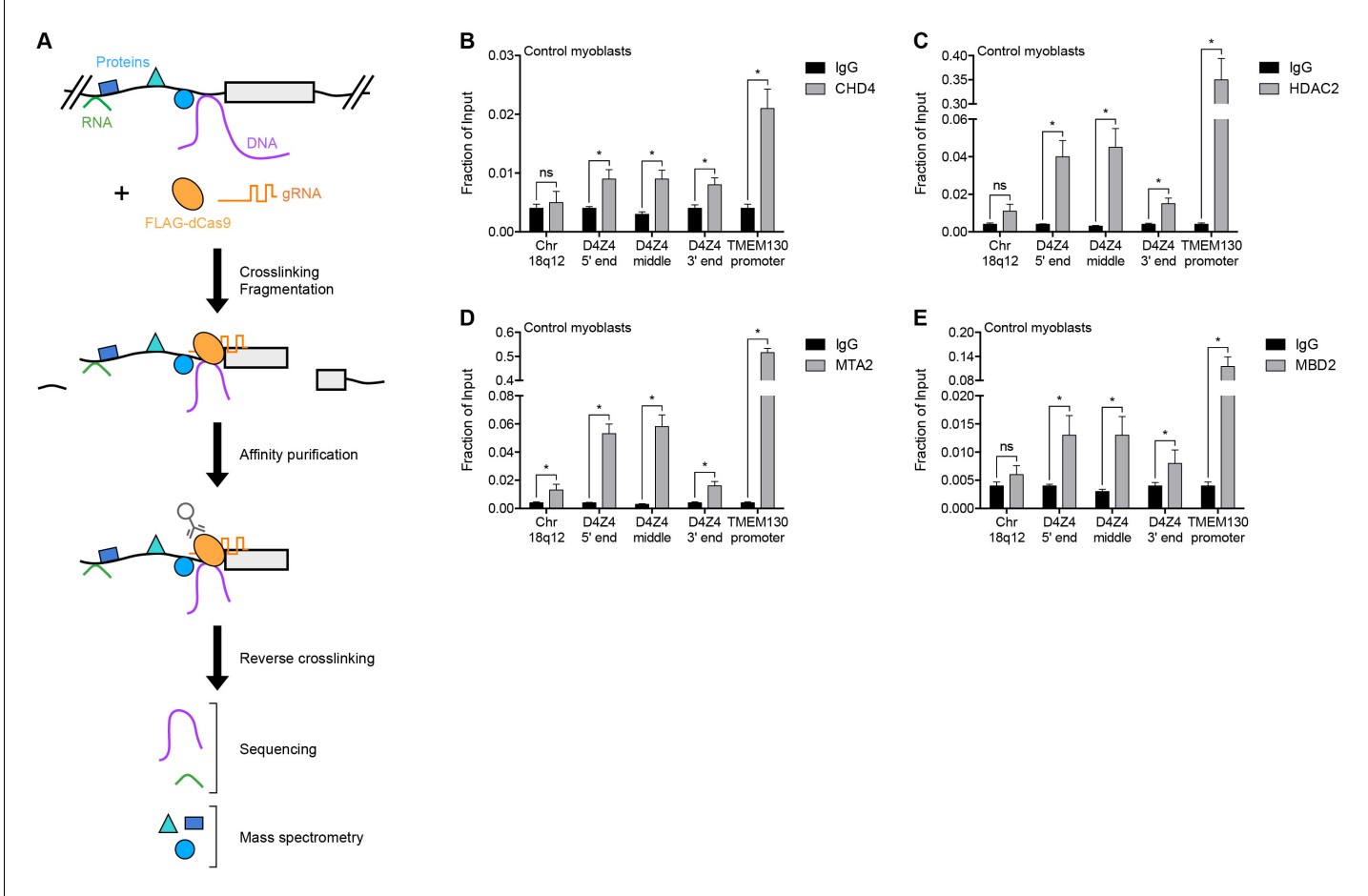

**Figure 1.** NuRD complex components bind the D4Z4 macrosatellite repeat. (**A**) Schematic summary of the enChIP procedure. A 3xFLAG-dCas9-HA-2xNLS fusion protein (FLAG-dCas9) consisting of an N-terminal triple FLAG (3xFLAG) epitope tag, catalytically inactive Cas9 endonuclease (dCas9), C-terminal human influenza hemagglutinin (HA) epitope tag and tandem nuclear localization signal (2xNLS) is expressed with one or more guide RNA (gRNA) in an appropriate cell context. Cells are crosslinked, chromatin is fragmented and complexes containing FLAG-dCas9 are immunoprecipitated with an anti-FLAG antibody. After reversing the crosslinks, molecules associated with the targeted genomic region are purified and identified by downstream analyses including mass spectrometry and next-generation sequencing. Adapted from *Fujita et al. (2016)*. (**B–E**) ChIP-qPCR enrichment of NuRD complex components CHD4 (**B**), HDAC2 (**C**), MTA2 (**D**) and MBD2 (**E**) along the D4Z4 repeat in MB2401 control myoblasts. The Chr18q12 amplicon contains no CpG dinucleotides and serves as a negative control site, while the TMEM130 promoter is occupied by NuRD complex components in published ENCODE datasets (*ENCODE Project Consortium, 2012*) and functions as a positive control locus. Error bars denote the standard deviation from the mean of three biological replicates. Statistical significance was calculated by comparing the specific pulldown to the IgG control at each site using a two-tailed, two-sample Mann-Whitney *U* test. *, $p \leq 0.05$; ns, not significant, $p > 0.05$. See also *Figure 1—source data 1*.
DOI: https://doi.org/10.7554/eLife.31023.003

The following source data and figure supplement are available for figure 1:

**Source data 1.** Source data for *Figure 1*.
DOI: https://doi.org/10.7554/eLife.31023.005
**Figure supplement 1.** Validation of myoblast cell lines used for enChIP-MS.
DOI: https://doi.org/10.7554/eLife.31023.004

and GATAD2A and GATAD2B (*Basta and Rauchman, 2015*) (*Figure 2A*). In MB2401 control myoblasts, small interfering RNA (siRNA) depletion of the lysine deacetylases HDAC1 or HDAC2 had no significant effect on DUX4 mRNA levels, whereas concurrent HDAC1/HDAC2 knockdown increased DUX4 mRNA 100-fold resulting in the activation of DUX4 target genes *ZSCAN4* and *TRIM43* (*Figure 2B* and *Figure 2—figure supplement 1A*). In contrast, in MB073 FSHD1 and MB200 FSHD2 myoblasts, singular HDAC1 or HDAC2 depletion led to a $\geq$ 20-fold activation of DUX4 mRNA while dual HDAC1/HDAC2 knockdown increased DUX4 levels more than 140-fold, with comparable

**Table 1.** Examples of proteins identified by enChIP-MS.

| Gene name | Sample | | | | Category |
| | gD4Z4 | | gMYOD1 | | |
| | # peptides* | % coverage† | # peptides* | % coverage† | |
|---|---|---|---|---|---|
| CBX3/HP1γ | 4.7 | 15.7 | 0.0 | 0.0 | Known D4Z4-associated proteins |
| NCL | 47.4 | 22.3 | 34.0 | 14.2 | |
| PDS5B | 2.0 | 11.2 | 0.0 | 0.0 | |
| RAD21 | 1.8 | 2.9 | 0.0 | 0.0 | |
| SMC1A | 7.0 | 5.8 | 2.0 | 1.8 | |
| SMC3 | 17.0 | 6.7 | 1.0 | 1.0 | |
| SMCHD1 | 1.6 | 2.4 | 0.0 | 0.0 | |
| CHD4 | 8.3 | 3.1 | 0.0 | 0.0 | NuRD complex components |
| HDAC2 | 2.5 | 5.6 | 3.0 | 6.8 | |
| MTA2 | 1.2 | 2.2 | 1.0 | 1.5 | |
| RBBP4 | 4.5 | 7.5 | 4.0 | 6.9 | |

*Average number of peptides recovered from each sample type, combining like (gD4Z4 or gMYOD1) immunoprecipitations.

†Average percentage of each protein covered by the identified peptides from each sample type, combining like (gD4Z4 or gMYOD1) immunoprecipitations.

DOI: https://doi.org/10.7554/eLife.31023.006

changes to DUX4 targets (*Figure 2C–D* and *Figure 2—figure supplement 1B–C*). Pharmacological inhibition of HDAC1/HDAC2 activity by MS-275 (*Nebbioso et al., 2009*) also increased DUX4 and DUX4 target gene expression, and enhanced histone H4 acetylation at the D4Z4 array (*Figure 2—figure supplement 2*). Collectively, these results indicate that HDAC1 and HDAC2 are associated with, and function to transcriptionally repress, the D4Z4 array. These data also show that the D4Z4 repeat in control myoblasts is more resistant to de-repression than the D4Z4 repeat in FSHD cells, which are sensitized because of a shortened array (FSHD1) or *SMCHD1* mutation (FSHD2).

We next evaluated the necessity of the ATP-dependent chromatin remodelers CHD3 and CHD4 for D4Z4 repeat repression. Depleting CHD4 from MB2401 control myoblasts had no effect on DUX4 expression (*Figure 2E* and *Figure 2—figure supplement 3A*). However, CHD4 knockdown in MB073 FSHD1 or MB200 FSHD2 myoblasts increased DUX4 mRNA 20-fold and concomitantly activated DUX4 targets (*Figure 2F–G* and *Figure 2—figure supplement 3B–C*). In contrast, CHD3 depletion did not lead to DUX4 de-repression in either control or FSHD cells (*Figure 2—figure supplement 4*), consistent with its absence from the gD4Z4 enChIP purifications and the mutually exclusive nature of CHD3 and CHD4 within the NuRD complex. Together, these results reveal that CHD4 binds the D4Z4 repeat and is necessary to silence DUX4 expression in FSHD cells, whereas control myoblasts have a more stably repressed D4Z4 array.

Similar to CHD4, depleting methyl-CpG-binding protein MBD2 from MB2401 control myoblasts had no effect on DUX4 mRNA levels (*Figure 2H* and *Figure 2—figure supplement 5A*). However, depleting MBD2 from MB073 FSHD1 myoblasts moderately, but significantly, increased DUX4 expression, whereas DUX4 was not de-repressed when MBD2 was knocked down in MB200 FSHD2 myoblasts (*Figure 2I–J* and *Figure 2—figure supplement 5B–C*). This difference suggests a possible D4Z4 context-dependent effect that was not observed for the single-copy NuRD complex-bound gene *TMEM130* following MBD2 knockdown (*Figure 2—figure supplement 5*). We further observed that depletion of MBD3, which can recruit the NuRD complex to unmethylated DNA (*Baubec et al., 2013*; *Le Guezennec et al., 2006*; *Saito and Ishikawa, 2002*), did not de-repress DUX4 in MB2401 control, MB073 FSHD1 or MB200 FSHD2 myoblasts (*Figure 2—figure supplement 6*). Together, these data show that MBD2 occupies the D4Z4 array and is necessary for DUX4 repression in at least some contexts, and suggest that factors in addition to MBD2 might recruit components shared by the NuRD complex to silence the D4Z4 macrosatellite repeat.

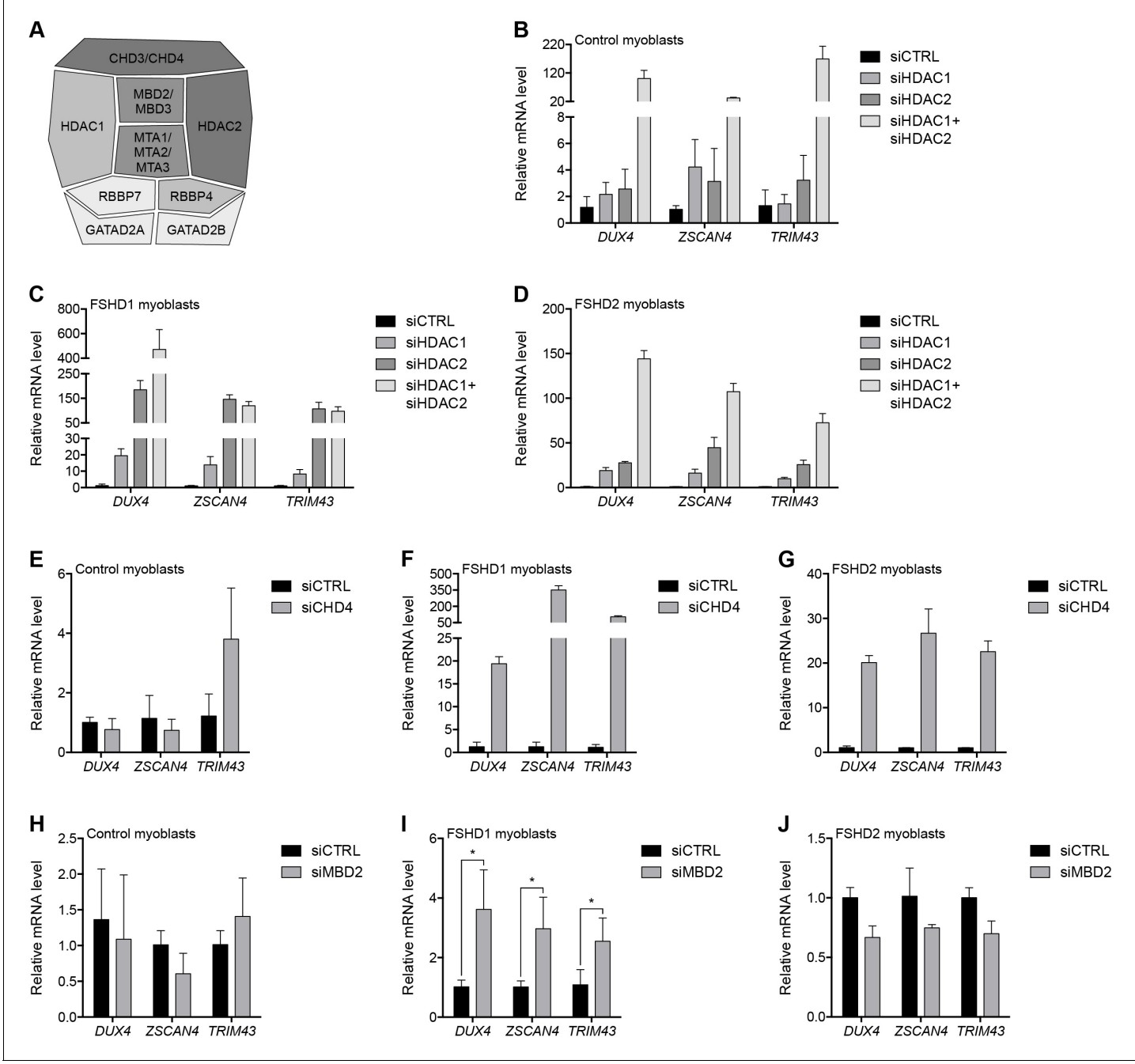

**Figure 2.** The MBD2/NuRD complex represses the D4Z4 array. (**A**) Schematic representation of the NuRD complex. Subunits colored darkest grey have the most lines of evidence linking them to DUX4 regulation (e.g. enChIP, siRNA and ChIP data), while more lightly colored subunits have progressively less experimental support. Adapted from *Hota and Bruneau (2016)*. (**B–J**) DUX4 and DUX4 target gene expression as determined by RT-qPCR following control (CTRL), HDAC1/HDAC2 (**B–D**), CHD4 (**E–G**) or MBD2 (**H–J**) siRNA knockdown in MB2401 control (**B,E,H**), MB073 FSHD1 (**C,F,I**) or MB200 FSHD2 (**D,G,J**) myoblasts. Error bars denote the standard deviation from the mean of three biological replicates. Statistical significance was calculated by comparing the specific knockdown to the control knockdown for each gene using a two-tailed, two-sample Mann-Whitney *U* test. *, p≤0.05.See also *Figure 2—source data 1*.

DOI: https://doi.org/10.7554/eLife.31023.007

The following source data and figure supplements are available for figure 2:

**Source data 1.** Source data for *Figure 2*.

DOI: https://doi.org/10.7554/eLife.31023.014

**Figure supplement 1.** Validation of HDAC1 and HDAC2 knockdown.

*Figure 2 continued on next page*

*Figure 2 continued*

DOI: https://doi.org/10.7554/eLife.31023.008

**Figure supplement 2.** Pharmacological inhibition of HDAC1/HDAC2.

DOI: https://doi.org/10.7554/eLife.31023.009

**Figure supplement 3.** Validation of CHD4 knockdown.

DOI: https://doi.org/10.7554/eLife.31023.010

**Figure supplement 4.** CHD3 depletion in control and FSHD myoblasts.

DOI: https://doi.org/10.7554/eLife.31023.011

**Figure supplement 5.** Validation of MBD2 knockdown.

DOI: https://doi.org/10.7554/eLife.31023.012

**Figure supplement 6.** MBD3 depletion in control and FSHD myoblasts.

DOI: https://doi.org/10.7554/eLife.31023.013

## Silencing the D4Z4 array requires components of the MBD1/CAF-1 complex

The NuRD complex is known to cooperate with other complexes to carry out its cellular functions. For example, NuRD and the CAF-1 chromatin assembly complex work together in several molecular processes (*Helbling Chadwick et al., 2009*; *Yang et al., 2015*) and share a core subunit, RBBP4, which was identified as associated with the D4Z4 repeat by gD4Z4 enChIP purification (*Table 1*). CHAF1A and CHAF1B comprise the other core members of the CAF-1 complex (*Volk and Crispino, 2015*) (*Figure 3A*). Depleting CHAF1A or CHAF1B resulted in the activation of DUX4 and DUX4 target genes in FSHD myoblasts (*Figure 3B–D* and *Figure 3—figure supplement 1*), confirming a role for this complex in D4Z4 regulation. CAF-1 interacts with CpG-binding protein MBD1, which binds both methylated and unmethylated DNA to inhibit transcription (*Jørgensen et al., 2004*; *Reese et al., 2003*). Knockdown of MBD1 led to DUX4 and DUX4 target gene activation in MB200 FSHD2 myoblasts but not in MB2401 control or MB073 FSHD1 myoblasts (*Figure 3E–G* and *Figure 3—figure supplement 2*), indicating a possible context-dependent relative necessity of MBD1 or MBD2 in different FSHD cells.

Notably, although knockdown of CHAF1A or CHD4 alone did not induce DUX4 expression in MB2401 control myoblasts (*Figure 2E* and *Figure 3B*), simultaneous depletion increased DUX4 mRNA levels over 150-fold (*Figure 3H* and *Figure 3—figure supplement 3A*). An additive or greater impact was also observed with dual versus singular CHD4 and CHAF1A knockdown in MB073 FSHD1 and MB200 FSHD2 myoblasts (*Figure 3I–J* and *Figure 3—figure supplement 3B–C*). Together, these results indicate that a combination of MBD1- and MBD2-mediated recruitment of the CAF-1 and NuRD repressive complexes, respectively, work together to silence the D4Z4 repeat in skeletal muscle cells.

To extend these studies, we depleted CHD4, CHAF1A, MBD2, or MBD1 in five additional FSHD cell lines: one FSHD1 cell line (54-2) with three 4qA D4Z4 repeats (compared to the 8 repeats of the MB073 line), and four FSHD2 lines (2305, 2453, 2338, and 1881) with different *SMCHD1* mutations and repeat sizes ranging from 11 to 15 D4Z4 units (*Supplementary file 3*). All five lines showed de-repression of DUX4 upon knockdown of MBD2 or CHAF1A, and all but one (2453, an FSHD2 cell line) showed increased DUX4 expression following CHD4 depletion, whereas de-repression following MBD1 knockdown was evident in the FSHD1 and two of the FSHD2 cell lines (*Figure 3—figure supplements 4–7*). Taken together, these data indicate the combined roles of the NuRD and CAF-1 complexes in repressing DUX4, and that the relative necessity of specific components of each pathway might vary depending on the cellular context, or possibly the efficiency of each knockdown.

## Components shared by the NuRD and CAF-1 complexes mediate D4Z4 repeat repression

To repress transcription, core members of the NuRD and CAF-1 complexes utilize a shared set of auxiliary factors, namely the tripartite motif-containing protein TRIM28, the lysine methyltransferase SETDB1, and the lysine demethylase KDM1A (*Ivanov et al., 2007*; *Loyola et al., 2009*; *Sarraf and Stancheva, 2004*; *Schultz et al., 2001*; *Wang et al., 2009*; *Yang et al., 2015*). Knockdown of TRIM28, SETDB1 or KDM1A de-repressed DUX4 in MB073 FSHD1 and MB200 FSHD2 myoblasts to

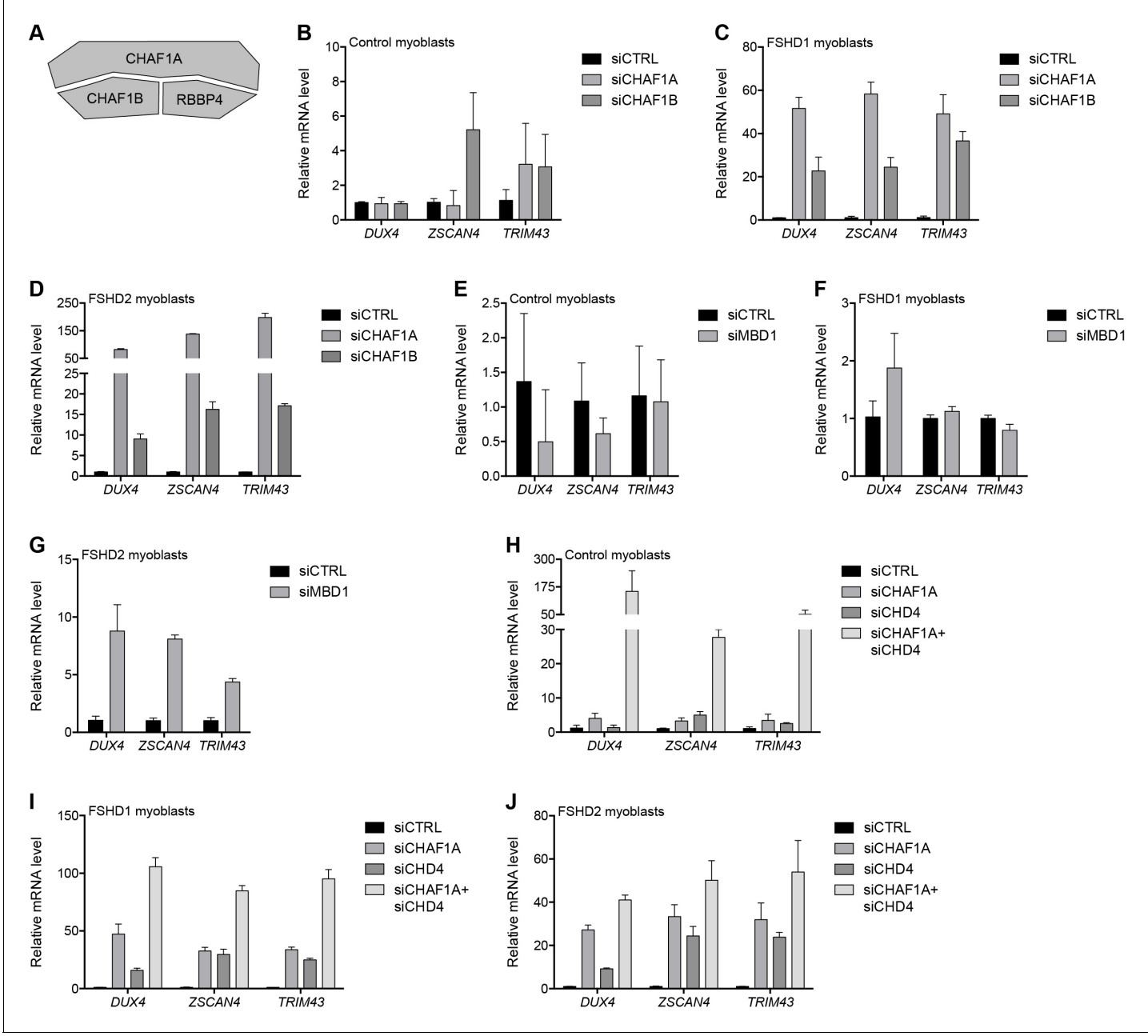

**Figure 3.** MBD1/CAF-1 components repress the D4Z4 array. (**A**) Schematic representation of the CAF-1 complex, with shading as in *Figure 2A*. All subunits have one line of evidence linking them to DUX4 regulation. (**B–J**) DUX4 and DUX4 target gene expression as determined by RT-qPCR following control (CTRL), CHAF1A/CHAF1B (**B–D**), MBD1 (**E–G**) or CHAF1A/CHD4 (**H–J**) siRNA knockdown in MB2401 control (**B,E,H**), MB073 FSHD1 (**C, F,I**) or MB200 FSHD2 (**D,G,J**) myoblasts. Error bars denote the standard deviation from the mean of three biological replicates. See also *Figure 3— source data 1*.

DOI: https://doi.org/10.7554/eLife.31023.015

The following source data and figure supplements are available for figure 3:

**Source data 1.** Source data for *Figure 3*.
DOI: https://doi.org/10.7554/eLife.31023.023

**Figure supplement 1.** Validation of CHAFA1 and CHAF1B knockdown.
DOI: https://doi.org/10.7554/eLife.31023.016

**Figure supplement 2.** Validation of MBD1 knockdown.
DOI: https://doi.org/10.7554/eLife.31023.017

**Figure supplement 3.** Validation of CHAFA1 and CHD4 knockdown.
DOI: https://doi.org/10.7554/eLife.31023.018

*Figure 3 continued on next page*

*Figure 3 continued*

DOI: https://doi.org/10.7554/eLife.31023.018

**Figure supplement 4.** CHD4 depletion in additional FSHD cell lines.

DOI: https://doi.org/10.7554/eLife.31023.019

**Figure supplement 5.** MBD2 depletion in additional FSHD cell lines.

DOI: https://doi.org/10.7554/eLife.31023.020

**Figure supplement 6.** CHAF1A depletion in additional FSHD cell lines.

DOI: https://doi.org/10.7554/eLife.31023.021

**Figure supplement 7.** MBD1 depletion in additional FSHD cell lines.

DOI: https://doi.org/10.7554/eLife.31023.022

varying degrees ranging from 3- to 130-fold (*Figure 4* and *Figure 4—figure supplements 1–3*), implicating them in facilitating silencing of the D4Z4 array. Of these factors, only KDM1A knockdown de-repressed DUX4 mRNA in the MB2401 control myoblasts, indicating a necessary role for this demethylase in maintaining repression of both normal and pathological D4Z4 alleles in muscle cells. In support of these expression data, peptides for TRIM28 were present in gD4Z4 enChIP pulldowns, although they did not meet our filtering criteria to be included in the list of D4Z4-associated proteins.

Similarly, SIN3A peptides were found in a gD4Z4 pulldown before our final filtering steps. The transcriptionally repressive SIN3 complex shares core proteins HDAC1, HDAC2, RBBP4, and RBBP7 with the NuRD complex and is also composed of SDS3, SAP18, SAP30 and SIN3A or SIN3B subunits (*Grzenda et al., 2009*) (*Figure 4—figure supplement 4A*). Therefore, we tested its role in D4Z4 repeat repression and found that SIN3A or SIN3B depletion led to the activation of DUX4 and DUX4 target genes in FSHD cells (*Figure 4—figure supplement 4B–G*), supporting a role for the SIN3 complex in D4Z4 regulation. Taken together, these data indicate that D4Z4 array silencing is mediated by multiple chromatin regulatory factors that act together with core components of the NuRD complex and also depend on the CAF-1 chromatin assembly complex to achieve full epigenetic repression.

## Proteins that repress the D4Z4 array in myoblasts also silence DUX4 in iPS cells

We previously reported that DUX4 is expressed at very low levels in human iPS cell populations (*Snider et al., 2010*) and, similar to the expression pattern in FSHD myoblasts, this represents the occasional expression in a small number of cells (JWL, unpublished data). We have more recently shown that DUX4 is present in four-cell human embryos and that when expressed in iPS cells or muscle cells it activates a cleavage-stage transcriptional program similar to the program expressed in a subset of 'naïve' iPS or embryonic stem (ES) cells (*Hendrickson et al., 2017*; *Whiddon et al., 2017*). To determine whether factors responsible for silencing the D4Z4 repeat in myoblasts have a similar function in a model of early development, we knocked down components of the NuRD and CAF-1 complexes in human eMHF2 iPS cells, which were derived from an unaffected (non-FSHD) individual, and assessed the impact on DUX4 expression. Similar to our myoblast results, depletion of HDAC1/HDAC2, CHD4, CHAF1A, SETDB1 or SIN3B de-repressed DUX4 in iPS cells; whereas, unlike in myoblasts, knockdown of KDM1A in iPS cells had a more minor effect on the levels of DUX4 mRNA (*Figure 5* and *Figure 5—figure supplement 1*).

To determine whether iPS cells have a greater necessity for NuRD and CAF-1 components to maintain DUX4 repression compared to somatic cells, we transduced a human foreskin fibroblast cell line (HFF3) with the reprogramming factors Oct4, Sox2, Nanog, and Lin28 to generate isogenic iPS cell clones (*Figure 5—figure supplement 2*). Notably, depletion of NuRD and CAF-1 complex components did not lead to DUX4 de-repression in the parental HFF3 fibroblast line, whereas the HFF3 iPS lines responded similarly to the eMHF2 iPS line (*Figure 5—figure supplement 3*). These results indicate that the NuRD and CAF-1 complexes that silence the D4Z4 macrosatellite array in muscle cells also contribute to the regulation of this locus in human iPS cells, and that iPS cells have decreased D4Z4 repression compared to their somatic counterpart, similar to the decreased repression in FSHD myoblasts compared to control myoblasts.

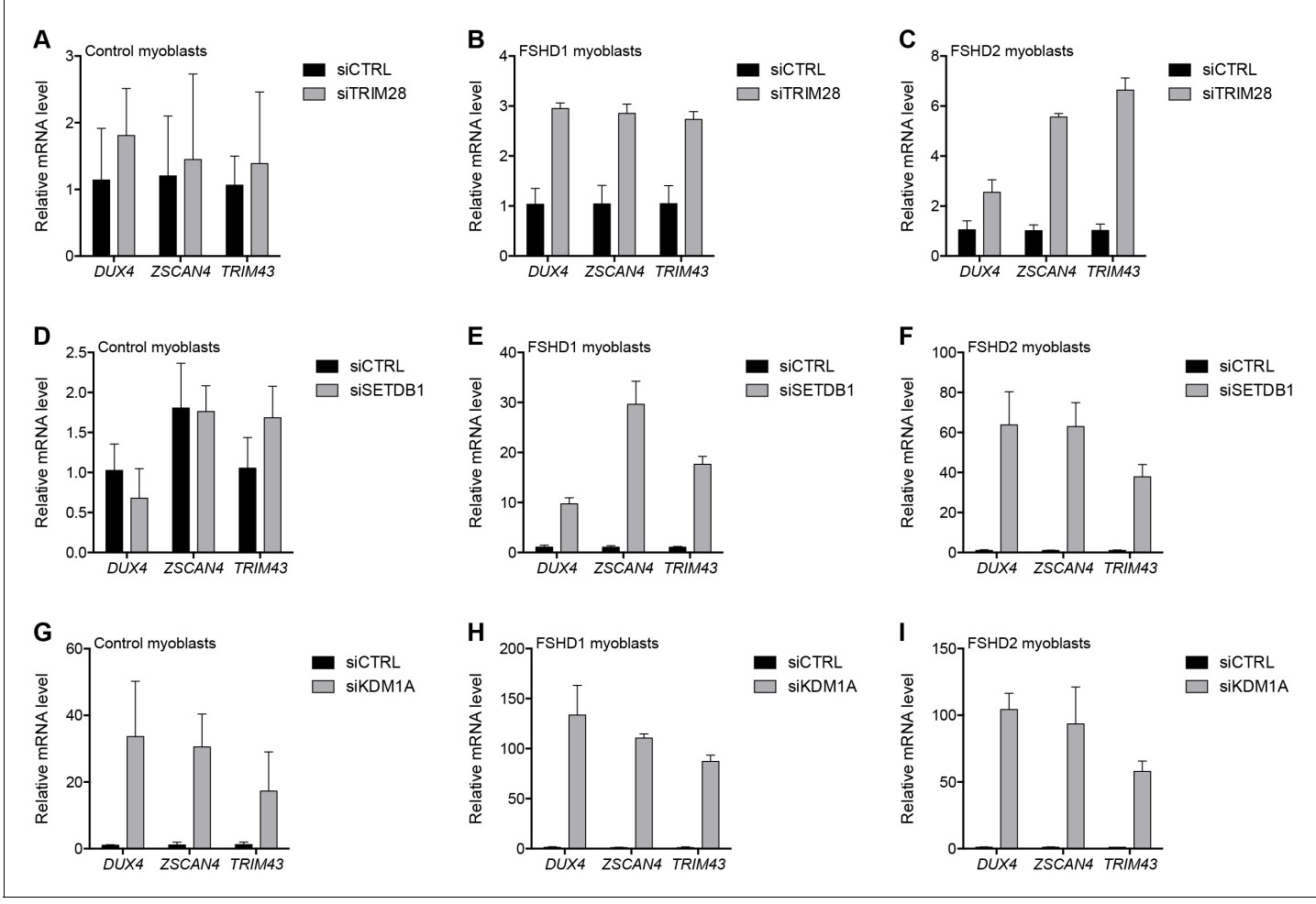

**Figure 4.** Additional transcriptional repressors silence the D4Z4 repeat. (A–I) DUX4 and DUX4 target gene expression as determined by RT-qPCR following control (CTRL), TRIM28 (A–C), SETDB1 (D–F) or KDM1A (G–I) siRNA knockdown in MB2401 control (A,D,G), MB073 FSHD1 (B,E,H) or MB200 FSHD2 (C,F,I) myoblasts. Error bars denote the standard deviation from the mean of three biological replicates. Statistical significance was calculated by comparing the specific knockdown to the control knockdown for each gene using a two-tailed, two-sample Mann-Whitney *U* test and p was ≤0.05 for all comparisons except those in (A) and (D). See also *Figure 4—source data 1*.

DOI: https://doi.org/10.7554/eLife.31023.024

The following source data and figure supplements are available for figure 4:

**Source data 1.** Source data for *Figure 4*.
DOI: https://doi.org/10.7554/eLife.31023.029
**Figure supplement 1.** Validation of TRIM28 knockdown.
DOI: https://doi.org/10.7554/eLife.31023.025
**Figure supplement 2.** Validation of SETDB1 knockdown.
DOI: https://doi.org/10.7554/eLife.31023.026
**Figure supplement 3.** Validation of KDM1A knockdown.
DOI: https://doi.org/10.7554/eLife.31023.027
**Figure supplement 4.** SIN3A/SIN3B knockdown.
DOI: https://doi.org/10.7554/eLife.31023.028

## MBD3L2 de-represses the D4Z4 repeat

In prior studies of DUX4-induced gene expression, we identified the MBD3L family (MBD3L2, MBD3L3, MBD3L4, and MBD3L5) as a direct target of DUX4 that was expressed in FSHD, but not control, muscle cells and muscle biopsies, and activated by exogenous DUX4 in cultured human myoblasts (*Geng et al., 2012*; *Yao et al., 2014*; *Young et al., 2013*). MBD3L family proteins can

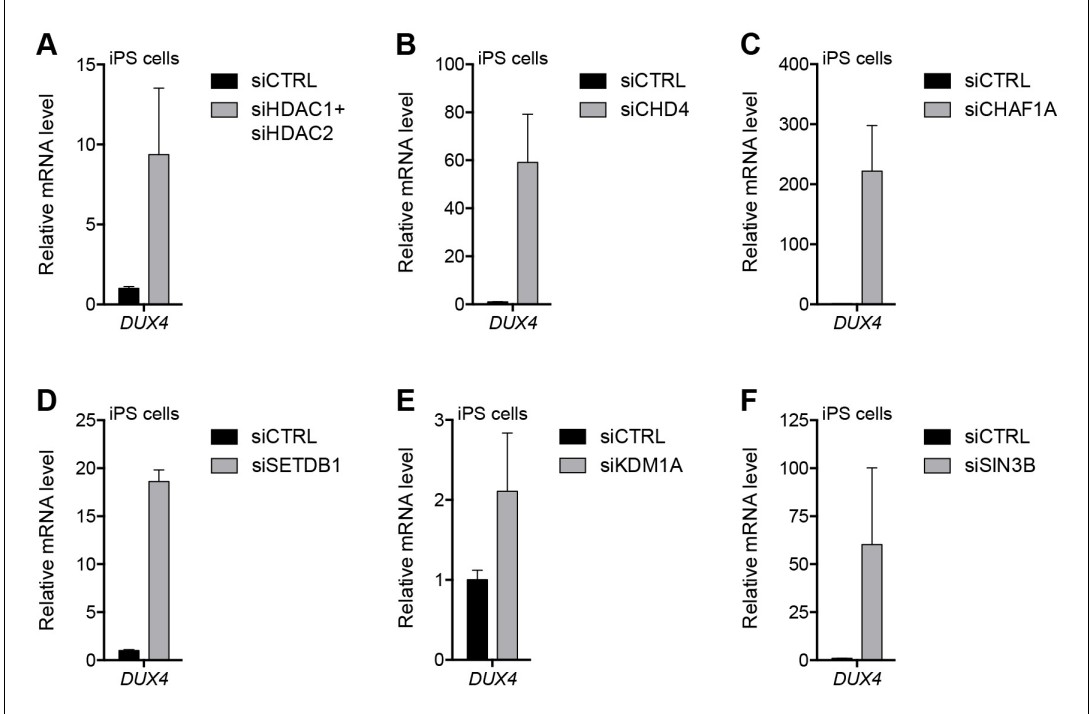

**Figure 5.** NuRD and CAF-1 complex components repress DUX4 in iPS cells. (A–F) DUX4 gene expression as determined by RT-qPCR in human eMHF2 iPS cells following control (CTRL), HDAC1/HDAC2 (A), CHD4 (B), CHAF1A (C), SETDB1 (D), KDM1A (E) or SIN3B (F) siRNA knockdown. Error bars denote the standard deviation from the mean of three biological replicates. Statistical significance was calculated by comparing the specific knockdown to the control knockdown for each gene using a two-tailed, two-sample Mann-Whitney *U* test and p was ≤0.05 unless otherwise specified as not significant (ns). See also *Figure 5—source data 1*.
DOI: https://doi.org/10.7554/eLife.31023.030

The following source data and figure supplements are available for figure 5:

**Source data 1.** Source data for *Figure 5*.
DOI: https://doi.org/10.7554/eLife.31023.034
**Figure supplement 1.** Validation of repressor protein knockdowns in eMHF2 iPS cells.
DOI: https://doi.org/10.7554/eLife.31023.031
**Figure supplement 2.** Validation of HFF3 iPS cell generation.
DOI: https://doi.org/10.7554/eLife.31023.032
**Figure supplement 3.** NuRD and CAF-1 knockdown in HFF3 fibroblasts and iPS cells.
DOI: https://doi.org/10.7554/eLife.31023.033

replace MBD2 or MBD3 in the NuRD complex but they lack the CpG-binding domain and antagonize NuRD-mediated transcriptional repression, possibly by preventing the complex from being recruited to its DNA targets (*Jiang et al., 2002*; *Jin et al., 2005*). To determine whether MBD3L proteins de-repress the NuRD complex-regulated D4Z4 array, we transduced control and FSHD myoblasts with a lentiviral vector delivering a doxycycline-inducible MBD3L2 transgene and, after selecting for transgene-expressing cells, analyzed DUX4 mRNA and protein after 48 hr of doxycycline treatment. Similar to the knockdown of NuRD complex members, expression of MBD3L2 induced DUX4 5–18-fold in MB073 FSHD1 and MB200 FSHD2 myoblasts and increased by 10-fold the number of myoblast nuclei expressing DUX4 protein, whereas DUX4 was not de-repressed in MB2401 control myoblasts (*Figure 6A–E* and *Figure 6—figure supplement 1A–C*).

When cultured in low mitogen differentiation media, myoblasts fuse to form multinucleated myotubes, and DUX4 expression increases in FSHD myotubes compared to myoblasts (*Balog et al., 2015*). To determine whether the DUX4-induced MBD3L proteins might contribute to the increased DUX4 expression in myotubes, we expressed short hairpin RNA (shRNA) to inhibit MBD3L RNAs in MB073 FSHD1 and MB200 FSHD2 myotubes and found that these decreased DUX4 and DUX4 target gene expression by ~50% and~30%, respectively (*Figure 6F–G*, *Figure 6—figure supplement*

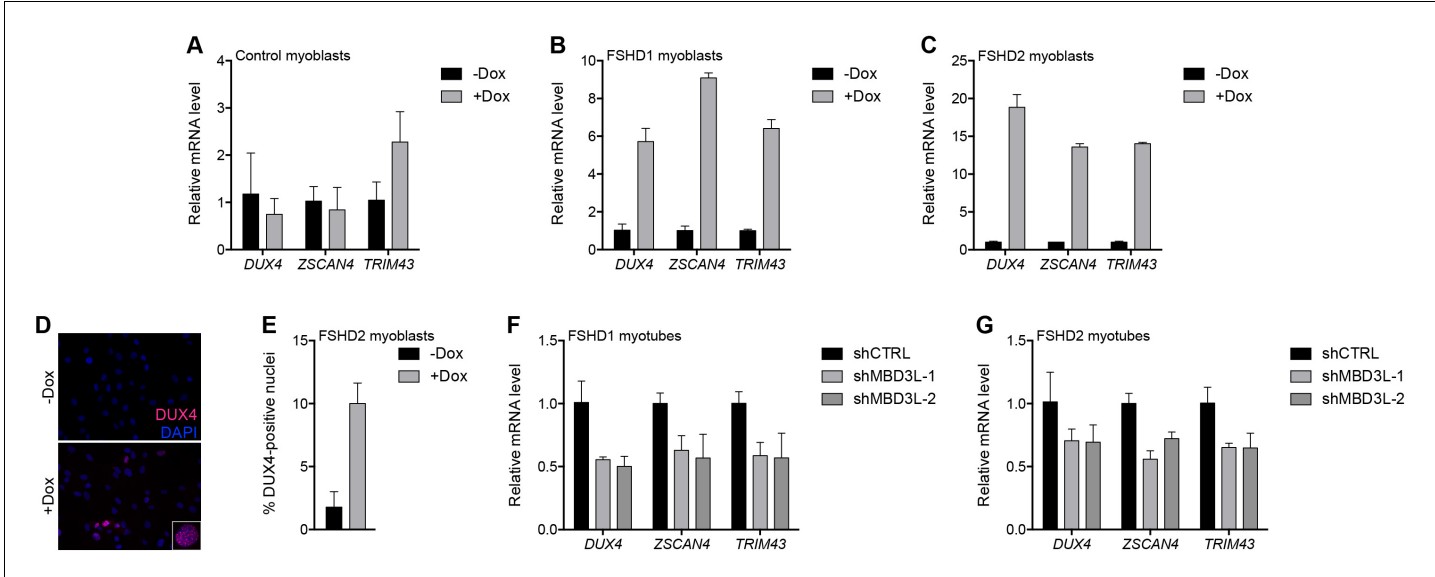

**Figure 6.** MBD3L2 expression de-represses the D4Z4 array. (**A–C**) DUX4 and DUX4 target gene expression as determined by RT-qPCR in MB2401 control (**A**), MB073 FSHD1 (**B**) or MB200 FSHD2 (**C**) myoblasts without (-) or with (+) doxycycline (Dox) treatment for 48 hr to induce *MBD3L2* transgene expression in clonal cell lines. (**D–E**) DUX4-positive nuclei upon overexpression of *MBD3L2* in MB200 FSHD2 myoblasts as in (**C**) were detected by immunofluorescence (**D**) and quantified by counting three fields representing >125 nuclei (**E**). (**F–G**) DUX4 and DUX4 target gene expression as determined by RT-qPCR following control (CTRL) or MBD3L family gene shRNA knockdown in MB073 FSHD1 (**F**) or MB200 FSHD2 (**G**) myotubes. Error bars denote the standard deviation from the mean of three biological replicates. Statistical significance was calculated by comparing the specific knockdown to the control knockdown for each gene using a two-tailed, two-sample Mann-Whitney *U* test and p was ≤0.05 for all comparisons except in (**A**). See also *Figure 6—source data 1*.

DOI: https://doi.org/10.7554/eLife.31023.035

The following source data and figure supplements are available for figure 6:

**Source data 1.** Source data for *Figure 6*.

DOI: https://doi.org/10.7554/eLife.31023.038

**Figure supplement 1.** Validation of MBD3L2 overexpression and depletion.

DOI: https://doi.org/10.7554/eLife.31023.036

**Figure supplement 2.** Additional MBD3L knockdown experiments.

DOI: https://doi.org/10.7554/eLife.31023.037

*1D–E* and *Figure 6—figure supplement 2*). Together, these data implicate MBD3L2 in the regulation of the D4Z4 array and demonstrate that endogenous DUX4-induced MBD3L proteins contribute to the amplification of DUX4 expression in FSHD myotubes.

## Discussion

In this study, enChIP-MS identified factors that co-purified with the D4Z4 macrosatellite array in human myoblasts, and subsequent ChIP and knockdown studies revealed that the NuRD and CAF-1 complexes repress DUX4 expression from the D4Z4 repeat in skeletal muscle and iPS cells. To some extent, each complex appears to have a parallel, or redundant, function in DUX4 repression because knockdown of both pathways was necessary to induce DUX4 expression in MB2401 control myoblasts. The distinctive mutations causing FSHD, or other factors such as the distribution of DNA methylation on the D4Z4, might preferentially weaken different specific components of each pathway, as evidenced by the relative necessity for CHD4, MBD2 or MBD1 in different FSHD cell lines. However, the variable efficiencies of the individual knockdowns in each cell type and experiment might also contribute to these apparent differences. It is also important to note that CAF-1 is a chromatin assembly complex and that the knockdowns were performed in replicating myoblasts; therefore, CAF-1 knockdown might not have the same consequence in post-mitotic myotubes. Overall, despite the relative differences in the necessity of the specific protein knockdown of individual

components of the NuRD or CAF-1 complexes in different FSHD cells, the data show that these complexes together are necessary to maintain D4Z4 repression. These two complexes also have shared auxiliary components, for example, TRIM28, SETDB1, and KDM1A, and knockdown of these factors also induced DUX4 expression in FSHD cells, with KDM1A knockdown being sufficient on its own to induce DUX4 in control myoblasts.

Our en-ChIP pulldowns identified several D4Z4-associated proteins that are involved in epigenetic silencing of variegated gene expression in mice (*Blewitt et al., 2005*; *Daxinger et al., 2013*). One of this group of *Modifier of murine metastable epiallele* (*Momme*) genes, Smchd1, was shown to directly repress DUX4 in human cells and to be a causative gene for FSHD2 (*Lemmers et al., 2012*). In addition to finding SMCHD1 associated with the D4Z4 in our enChIP, we identified the *Momme* genes PBRM1, RIF1, SMARCA4, SMARCA5 and UHRF1 as D4Z4-associated by enChIP-MS, and implicated the *Momme* genes HDAC1, SETDB1 and TRIM28 in the regulation of DUX4 through knockdown experiments. The convergence and striking overlap of the results of these two complementary approaches to understanding variegated gene expression suggest that conserved machinery may be responsible for repressing this type of locus across species. The presence of chromatin remodelers and positive transcriptional regulators, such as SMARCA5, BRD3 and BRD4, at the D4Z4 locus in the control cells used for the enChIP also indicates a dynamic balance between activators and repressors, which is consistent with the identification of sense and anti-sense transcripts associated with the D4Z4 repeats in both control and FSHD cells (*Snider et al., 2010*). Our findings also suggest that PBRM1, RIF1, SMARCA4, SMARCA5 and UHRF1 are candidates for playing a role in DUX4 regulation and deserve additional attention in future studies.

The necessity of the NuRD complex to maintain repression of DUX4 in FSHD cells suggests that the DUX4-mediated induction of the MBD3L family of factors might amplify DUX4 expression within a nucleus or facilitate the internuclear spreading of DUX4 in multinucleated myotubes. MBD3L factors replace MBD2 or MBD3 in the NuRD complex and antagonize its normal repressive function (*Jiang et al., 2002*; *Jin et al., 2005*). In this study, we showed that expression of MBD3L2 was sufficient to amplify DUX4 expression in FSHD cells and knockdown using an shRNA that targets the entire family showed that expression of the MBD3L family was necessary for the full induction of DUX4 expression in FSHD myotubes. The fact that DUX4 induces high expression of the clustered MBD3L genes reveals a positive feed-forward mechanism that might facilitate spreading of DUX4 expression between nuclei in myotubes. In FSHD myotubes, DUX4 expression apparently initiates in a single nucleus and the protein then spreads to adjacent nuclei in the syncytium. Similarly, MBD3L proteins are detected as spreading to adjacent nuclei (AEC, unpublished data) where they would facilitate DUX4 expression. In this manner, each DUX4 expressing nucleus would act to progressively amplify DUX4 expression in its neighbors, spreading DUX4 expression along the myofiber. This might be similar to, and additive with, the prior observation that the DUX4-mediated inhibition of NMD can amplify DUX4 expression by stabilizing the DUX4 mRNA, which is itself a target of NMD (*Feng et al., 2015*). It is interesting to speculate that the internuclear amplification of DUX4 expression might contribute to the susceptibility of skeletal muscle to damage in FSHD.

Together our data provide several complementary approaches to the challenge of creating an FSHD therapeutic. One strategy would be to enhance D4Z4 repression by designing drugs that increase NuRD complex-mediated repression. Although drugs that decrease epigenetic repression are in clinical use, including some that target members of the NuRD complex (SAHA, targeting HDAC1/2; ORY-1001, targeting KDM1A; GSK126, EZH2 inhibitor) drugs that enhance epigenetic repression have received less attention. This is partly due to concerns that they might also suppress important tumor suppressor genes, but the fact that mutations in *SMCHD1* and *DNMT3B* that cause FSHD have limited genome-wide consequences suggests that some factors might be relatively specific for repressing repetitive regions of the genome. A second strategy would be to prevent the amplification of DUX4 after it stochastically 'bursts' on in a myotube nucleus. This might be accomplished by inhibiting the production of MBD3L proteins with small molecules or interfering RNAs. Alternatively, myoblast transplantation with cells containing larger D4Z4 repeat sizes or 4qB alleles might provide 'decoy' nuclei that would absorb MBD3L factors and not activate DUX4, or, in a similar decoy approach, autologous transplants following deletion of the D4Z4 array and/or the MBD3L cluster.

Although little is known about the regulation of DUX4 expression in cleavage-stage embryos and the testis luminal cells, it is evident from this study that the expression of DUX4 in a small percentage

of iPS cells or ES cells shares mechanisms of molecular regulation with skeletal muscle cells. This also indicates similarities between the regulation of the human DUX4 retrogene and the mouse Dux retrogene that is also in a macrosatellite array, although thought to have arisen from a separate retrotransposition of the DUXC gene (*Clapp et al., 2007*; *Leidenroth et al., 2012*; *Leidenroth and Hewitt, 2010*). It was previously shown that CAF-1 depletion in mouse ES cells resulted in the expression of genes specific to two-cell embryos (*Ishiuchi et al., 2015*), and later shown that induction of these genes was blocked by simultaneous knockdown of mouse Dux along with Chaf1a (*Hendrickson et al., 2017*). Trim28, Kdm1a, and HDAC inhibitors have been shown to regulate Zscan4 and the early cleavage program in mouse ES cells (*Macfarlan et al., 2012*), and for Trim28 this activity was shown to be mediated through the induction of mouse Dux (*De Iaco et al., 2017*). Similarly, the NuRD complex and MBD3 have been shown to inhibit cellular reprogramming in mouse ES cells, and, conversely, reprogramming to a naive stem cell state was facilitated by inhibition of these complexes (*Luo et al., 2013*; *Rais et al., 2013*). The fact that inhibiting NuRD or CAF-1 activity potentiates stem cell reprogramming in mouse ES/iPS cells and, as shown in this report, potentiates human DUX4 expression, suggests that DUX4 itself might facilitate reprogramming to the naive state and that mouse Dux and human DUX4 might be subject to similar regulation, a finding not entirely obvious given that these retrogenes are thought to have been generated by independent retrotranspositions of the parental DUXC gene, as noted above.

In summary, we identified components of the NuRD and CAF-1 complexes as necessary to maintain repression of DUX4 expression from the D4Z4 repeat. In control myoblasts, either pathway was sufficient to maintain repression of DUX4, whereas in FSHD cells inhibition of either pathway resulted in higher levels of DUX4 expression. These same mechanisms repress DUX4 expression in iPS cells. In addition, the DUX4 induction of the NuRD antagonist MBD3L family further de-repressed DUX4 in FSHD cells. Together, these findings provide the basis for therapies directed at repressing DUX4 in FSHD and reveal a mechanism for the regulation of DUX4 in stem cells.

# Materials and methods

**Key resources table**

| Reagent type (species) or resource | Designation | Source or reference | Identifiers | Additional information |
|---|---|---|---|---|
| Cell line (*H. sapiens*) | 1881 myoblasts | Fields Center for FSHD and Neuromuscular Research at the University of Rochester Medical Center (https://www.urmc.rochester.edu/neurology/fields-center.aspx) | | See *Supplementary file 3* |
| Cell line (*H. sapiens*) | 2305 myoblasts | Fields Center for FSHD and Neuromuscular Research at the University of Rochester Medical Center (https://www.urmc.rochester.edu/neurology/fields-center.aspx) | | See *Supplementary file 3* |
| Cell line (*H. sapiens*) | 2338 myoblasts | Fields Center for FSHD and Neuromuscular Research at the University of Rochester Medical Center (https://www.urmc.rochester.edu/neurology/fields-center.aspx) | | See *Supplementary file 3* |
| Cell line (*H. sapiens*) | 2453 myoblasts | Fields Center for FSHD and Neuromuscular Research at the University of Rochester Medical Center (https://www.urmc.rochester.edu/neurology/fields-center.aspx) | | See *Supplementary file 3* |
| Cell line (*H. sapiens*) | 54–2 myoblasts | (*Krom et al., 2012*) (DOI: 10.1016/j.ajpath.2012.07.007) | | See *Supplementary file 3* |

*Continued on next page*

*Continued*

| Reagent type (species) or resource | Designation | Source or reference | Identifiers | Additional information |
|---|---|---|---|---|
| Cell line (*H. sapiens*) | eMHF2 iPS cells | University of Washington Institute for Stem Cell and Regenerative Medicine Tom and Sue Ellison Stem Cell Core (http://depts.washington.edu/iscrm/ellison) | | |
| Cell line (*H. sapiens*) | HFF3 fibroblasts | ATCC | ATCC:SCRC-1043; RRID:CVCL_DB29 | |
| Cell line (*H. sapiens*) | MB073 myoblasts | Fields Center for FSHD and Neuromuscular Research at the University of Rochester Medical Center (https://www.urmc.rochester.edu/neurology/fields-center.aspx) | | See *Supplementary file 3* |
| Cell line (*Homo sapiens*) | MB135 myoblasts | Fields Center for FSHD and Neuromuscular Research at the University of Rochester Medical Center (https://www.urmc.rochester.edu/neurology/fields-center.aspx) | | See *Supplementary file 3* |
| Cell line (*H. sapiens*) | MB200 myoblasts | Fields Center for FSHD and Neuromuscular Research at the University of Rochester Medical Center (https://www.urmc.rochester.edu/neurology/fields-center.aspx) | | See *Supplementary file 3* |
| Cell line (*H. sapiens*) | MB2401 myoblasts | Fields Center for FSHD and Neuromuscular Research at the University of Rochester Medical Center (https://www.urmc.rochester.edu/neurology/fields-center.aspx) | | See *Supplementary file 3* |
| Antibody | alpha-Tubulin | Sigma-Aldrich | Sigma-Aldrich:T9026; RRID:AB_477593 | |
| Antibody | Acetyl-Histone H4 | EMD Millipore | EMD Millipore:06866; RRID:AB_310270 | |
| Antibody | CHD4 | Bethyl Laboratories | Bethyl Laboratories: A301081A; RRID:AB_873001 | |
| Antibody | DUX4 (14–3) | (*Geng et al., 2011*) | | |
| Antibody | DUX4 (E5-5) | (*Geng et al., 2011*) | | |
| Antibody | FITC anti-mouse | Jackson ImmunoResearch | Jackson Immuno Research:715095150; RRID:AB_2340792 | |
| Antibody | FLAG M2 | Sigma-Aldrich | Sigma-Aldrich:F1804 or F3165; RRID:AB_262044 or RRID:AB_259529 | |
| Antibody | HDAC2 | Abcam | Abcam:ab7029; RRID:AB_305706 | |
| Antibody | HRP anti-mouse | Jackson ImmunoResearch | Jackson ImmunoResearch: 115035146; RRID:AB_2307392 | |
| Antibody | MBD2 | Bethyl Laboratories | Bethyl Laboratories: A301632A; RRID:AB_1211478 | |
| Antibody | MTA2 | Abcam | Abcam:ab8106; RRID:AB_306276 | |

*Continued on next page*

*Continued*

| Reagent type (species) or resource | Designation | Source or reference | Identifiers | Additional information |
|---|---|---|---|---|
| Antibody | TRITC anti-rabbit | Jackson ImmunoResearch | Jackson Immuno Research:711025152; RRID:AB_2340588 | |
| Recombinant DNA reagent | pCW57.1-MBD3L2 | This paper | Addgene plasmid #106332 | Lentiviral vector expressing doxycycline-inducible MBD3L2 (generated using pCW57.1 [Addgene plasmid #41393]) |
| Recombinant DNA reagent | pGIPZ-shControl | Fred Hutchinson Cancer Research Center Genomics Shared Resource (http://monod.fhcrc.org/rnai/) | | |
| Recombinant DNA reagent | pGIPZ-shMBD3L-1 | Fred Hutchinson Cancer Research Center Genomics Shared Resource (http://monod.fhcrc.org/rnai/) | | |
| Recombinant DNA reagent | pGIPZ-shMBD3L-2 | Fred Hutchinson Cancer Research Center Genomics Shared Resource (http://monod.fhcrc.org/rnai/) | | |
| Recombinant DNA reagent | pZLCv2-3xFLAG-dCas 9-HA-2xNLS | This paper | Addgene plasmid #106357 | Lentiviral vector expressing FLAG-tagged, nuclease-deficient Cas9 (generated using lentiCRISPRv2 [Addgene plasmid #52961] and pHR-SFFV-KRAB-dCas9-P2A-mCherry [Addgene plasmid #60954]) |
| Recombinant DNA reagent | pZLCv2-gD4Z4-1-3x FLAG-dCas9-HA-2xNLS | This paper | Addgene plasmid #106352 | Lentiviral vector expressing FLAG-dCas9 and a guide RNA targeting the D4Z4 unit |
| Recombinant DNA reagent | pZLCv2-gD4Z4-2-3x FLAG-dCas9-HA-2xNLS | This paper | Addgene plasmid #106353 | Lentiviral vector expressing FLAG-dCas9 and a gRNA targeting the D4Z4 unit |
| Recombinant DNA reagent | pZLCv2-gD4Z4-3-3x FLAG-dCas9-HA-2xNLS | This paper | Addgene plasmid #106354 | Lentiviral vector expressing FLAG-dCas9 and a gRNA targeting the D4Z4 unit |
| Recombinant DNA reagent | pZLCv2-gMYOD1-3x FLAG-dCas9-HA-2xNLS | This paper | Addgene plasmid #106355 | Lentiviral vector expressing FLAG-dCas9 and a gRNA targeting the MYOD1 distal regulatory region |
| Sequenced-based reagent | enChIP-/ChIP-qPCR primers | This paper | | See *Supplementary file 4* |
| Sequenced-based reagent | gRNAs | This paper | | See *Supplementary file 4* |
| Sequenced-based reagent | RT-qPCR primers | This paper | | See *Supplementary file 4* |
| Sequenced-based reagent | shRNAs | This paper | | See *Supplementary file 4* |
| Sequenced-based reagent | siRNAs | This paper | | See *Supplementary file 4* |
| Peptide, recombinant protein | 3X FLAG peptide | Sigma-Aldrich | Sigma-Aldrich:F4799 | |
| Commercial assay or kit | QIAshredder | Qiagen | Qiagen:79656 | |
| Commercial assay or kit | RNeasy Mini Kit | Qiagen | Qiagen:74106 | |

*Continued on next page*

*Continued*

| Reagent type (species) or resource | Designation | Source or reference | Identifiers | Additional information |
|---|---|---|---|---|
| Commercial assay or kit | SuperScript III First-Strand Synthesis System | Invitrogen/Thermo Fisher | Invitrogen/Thermo Fisher:18080051 | |
| Chemical compound, drug | 2-Mercaptoethanol | Sigma-Aldrich | Sigma-Aldrich:M3148 | |
| Chemical compound, drug | Dexamethasone | Sigma-Aldrich | Sigma-Aldrich:D4902 | |
| Chemical compound, drug | DMEM:Nutrient Mixture F-12 | Gibco/Thermo Fisher | Gibco/Thermo Fisher:11320082 | |
| Chemical compound, drug | DNase I | Thermo Fisher | Thermo Fisher:18068015 | |
| Chemical compound, drug | Doxycyline hyclate | Sigma-Aldrich | Sigma-Aldrich:D9891 | |
| Chemical compound, drug | Dulbecco's Modified Eagle Medium (DMEM) | Gibco/Thermo Fisher | Gibco/Thermo Fisher:11965092 | |
| Chemical compound, drug | Dynabeads-Protein G | Thermo Fisher | Thermo Fisher:10003D | |
| Chemical compound, drug | Ham's F-10 Nutrient Mix | Gibco/Thermo Fisher | Gibco/Thermo Fisher:11550043 | |
| Chemical compound, drug | Horse serum | Gibco/Thermo Fisher | Gibco/Thermo Fisher:26050070 | |
| Chemical compound, drug | HyClone Fetal Bovine Serum | GE Healthcare Life Sciences | GE Healthcare Life Sciences:SH30071.03 | |
| Chemical compound, drug | Insulin | Sigma-Aldrich | Sigma-Aldrich:I1882 | |
| Chemical compound, drug | KnockOut Serum Replacement | Gibco/Thermo Fisher | Gibco/Thermo Fisher:10828028 | |
| Chemical compound, drug | Lipofectamine RNAiMAX | Invitrogen/Thermo Fisher | Invitrogen/Thermo Fisher:13778150 | |
| Chemical compound, drug | Matrigel | Corning Life Science | Corning Life Science:354277 | |
| Chemical compound, drug | MEM non-essential amino acids | Gibco/Thermo Fisher | Gibco/Thermo Fisher:11140050 | |
| Chemical compound, drug | mTeSR1 medium | STEMCELL Technologies | STEMCELL Technologies:85850 | |
| Chemical compound, drug | Opti-MEM reduced serum medium | Thermo Fisher | Thermo Fisher:31985070 | |
| Chemical compound, drug | Penicillin/streptomycin | Gibco/Thermo Fisher | Gibco/Thermo Fisher:15140122 | |
| Chemical compound, drug | Polybrene | Sigma-Aldrich | Sigma-Aldrich:107689 | |
| Chemical compound, drug | Puromycin | Sigma-Aldrich | Sigma-Aldrich:P8833 | |
| Chemical compound, drug | Recominant human basic fibroblast growth factor | Promega Corporation | Promega Corporation:G5071 | |
| Chemical compound, drug | Sodium pyruvate | Gibco/Thermo Fisher | Gibco/Thermo Fisher:11360070 | |
| Chemical compound, drug | Transferrin | Sigma-Aldrich | Sigma-Aldrich:T0665 | |
| Chemical compound, drug | Y-27632 ROCK inhibitor | Miltenyi Biotec | Miltenyi Biotec:130106538 | |

*Continued on next page*

*Continued*

| Reagent type (species) or resource | Designation | Source or reference | Identifiers | Additional information |
|---|---|---|---|---|
| Software, algorithm | Code used for proteomics data analysis | This paper (*Jagannathan, 2017*) | | The R code used for the proteomics data analysis can be accessed via github at https://github.com/sjaganna/2017-campbell_et_al |
| Software, algorithm | GraphPad Prism | GraphPad Prism (https://graphpad.com) | RRID:SCR_015807 | Version 6 |
| Software, algorithm | ImageJ | ImageJ (http://imagej.nih.gov/ij/) | RRID:SCR_003070 | |
| Software, algorithm | Proteome Discoverer | Thermo Fisher | RRID:SCR_014477 | Version 1.4 |

## Cell culture and reagents

All reagents were obtained from Sigma-Aldrich (St. Louis, MO) unless otherwise specified. Human primary myoblast cell lines originated from the Fields Center for FSHD and Neuromuscular Research at the University of Rochester Medical Center (https://www.urmc.rochester.edu/neurology/fields-center.aspx) and were immortalized by retroviral transduction of CDK4 and hTERT (*Stadler et al., 2011*). Myoblasts were maintained in Ham's F-10 Nutrient Mix (Gibco, Waltham, MA) supplemented with 20% HyClone Fetal Bovine Serum (GE Healthcare Life Sciences, Pittsburgh, PA), 100 U/100 μg penicillin/streptomycin (Gibco), 10 ng/ml recombinant human basic fibroblast growth factor (Promega Corporation, Madison, WI) and 1 μM dexamethasone. Differentiation of myoblasts into myotubes was achieved by switching the fully confluent myoblast monolayer into Dulbecco's Modified Eagle Medium (DMEM, Gibco) containing 1% horse serum (Gibco), 100 U/100 μg penicillin/streptomycin, 10 μg/ml insulin and 10 μg/ml transferrin for 48–72 hr. Myoblasts harboring a transgene were additionally cultured in 2 μg/ml puromycin and transgene expression induced with 1 μg/ml doxycycline hyclate when required. Myoblast cell line identity was authenticated by monitoring fusion into myotubes, DUX4 expression, and the presence of a 4qA161 allele. Detailed characteristics of the myoblast lines used in this study are provided in *Supplementary file 3*. Human control (non-FSHD) iPS cells were obtained from the University of Washington Institute for Stem Cell and Regenerative Medicine Tom and Sue Ellison Stem Cell Core (eMHF2) (*Hendrickson et al., 2017*) or derived in-house from normal HFF3 foreskin fibroblasts reprogrammed via lentiviral transduction of Oct4, Sox2, Nanog and Lin28 (*Yu et al., 2007*), and grown in DMEM:Nutrient Mixture F-12 (1:1, Gibco) with 100 U/100 μg penicillin/streptomycin, 10 mM MEM Non-Essential Amino Acids (Gibco), 100 mM sodium pyruvate (Thermo Fisher Scientific, Waltham, MA, USA), 20% KnockOut Serum Replacement (Gibco), 1 mM 2-mercaptoethanol and 4 ng/ml recombinant human basic fibroblast growth factor under hypoxic (5% $O_2$) conditions on 0.1% gelatin-coated plates pre-seeded with $1.3 \times 10^4$ cells/cm$^2$ of irradiated mouse embryonic fibroblasts. While the full haplotypes are unknown, eMHF2 cells utilize DUX4 exon 3, suggesting a 4qA161S allele, while HFF3 cells use DUX4 exon 3b, suggesting a 4qA161L allele (*Lemmers et al., 2018*). HFF3 fibroblasts and 293T cells were maintained in DMEM supplemented with 10% HyClone Fetal Bovine Serum and 100 U/100 μg penicillin/streptomycin. Cell lines are tested periodically for *Mycoplasma* contamination by the Fred Hutchinson Cancer Research Center Specimen Processing/Research Cell Bank and have not shown evidence of *Mycoplasma*.

## Cloning, virus production and transgenic cell line generation

To construct FLAG-dCas9-gRNA plasmids, the lentiCRISPRv2 vector (a gift from Feng Zhang, Addgene plasmid #52961) (*Sanjana et al., 2014*) was digested with AgeI and BamHI, PCR was used to amplify AgeI-3xFLAG-EcoRI from a synthesized template and EcoRI-dCas9-BamHI from pHR-SFFV-KRAB-dCas9-P2A-mCherry (a gift from Jonathan Weissman, Addgene plasmid #60954) (*Gilbert et al., 2014*), the three fragments were ligated together to create a 3xFLAG-dCas9-HA-2xNLS vector, and then D4Z4 or MYOD1 gRNA were inserted by digesting 3xFLAG-dCas9-HA-2xNLS with BsmBI and ligating it to annealed gRNA oligos. To construct the doxycycline-inducible MBD3L2 plasmid, the *MBD3L2* coding region was subcloned into the NheI and SalI sites of the pCW57.1 vector (a gift from David Root, Addgene plasmid #41393). The pGIPZ-shControl and -shMBD3L vectors were obtained from the Fred Hutchinson Cancer Research Center Genomics

Shared Resource. Lentiviral particles were produced in 293T cells by co-transfecting the appropriate lentiviral vector with pMD2.G (a gift from Didier Trono, Addgene plasmid #12259) and psPAX2 (a gift from Didier Trono, Addgene plasmid #12260) using Lipofectamine 2000 (Invitrogen, Carlsbad, CA) following the manufacturer's instructions. To generate polyclonal transgenic cell lines, myoblasts were transduced with lentivirus in the presence of 8 µg/ml polybrene and selected using 2 µg/ml puromycin. Monoclonal transgenic lines were generated by transducing at a low cell density using a low multiplicity of infection (MOI <1) and allowing cells that survived selection to form colonies before individual clones were isolated using cloning cylinders.

## Protein extraction and immunoblotting

Total protein extracts were generated by lysing cells in SDS sample buffer (500 mM Tris-HCl pH 6.8, 8% SDS, 20% 2-mercaptoethanol, 0.004% bromophenol blue, 30% glycerol) followed by sonication and boiling with 50 mM DTT. Samples were run on NuPage 4–12% precast polyacrylamide gels (Invitrogen) and transferred to nitrocellulose membrane (Invitrogen). Membranes were blocked in PBS containing 0.1% Tween-20% and 5% non-fat dry milk for 1 hr at room temperature before overnight incubation at 4°C with primary antibodies in block solution. Membranes were then incubated for 1 hr at room temperature with horseradish peroxidase-conjugated secondary antibodies in block solution and chemiluminescent substrate (Thermo Fisher Scientific) used for detection on film.

## Immunofluorescence

Cells were fixed in PBS containing 2% paraformaldehyde (Electron Microscopy Sciences, Hatfield, PA) for 7 min at room temperature and permeabilized for 10 min in PBS with 0.5% Triton X-100. Samples were then incubated overnight at 4°C with primary antibodies, followed by incubation with appropriate FITC- or TRITC-conjugated secondary antibodies for 1 hr at room temperature prior to DAPI counterstaining and imaging with a Zeiss Axiophot fluorescent microscope, AxioCam MRc digital camera and AxioVision 4.6 software (Carl Zeiss Microscopy, Thornwood, NY). Image J software (*Schneider et al., 2012*) was used for image analysis and quantification.

## enChIP-qPCR

FLAG-dCas9 chromatin occupancy was analyzed as previously described (*Fujita and Fujii, 2013*) using chromatin extraction and fragmentation methods from (*Forsberg et al., 2000*) and the following minor modifications. Five million trypsinized myoblasts were crosslinked with 1% formaldehyde (Thermo Fisher Scientific) for 10 min at room temperature. Chromatin was diluted to 0.5% SDS with IP Dilution Buffer (20 mM Tris pH 8.0, 2 mM EDTA, 150 mM NaCl, 1% Triton X-100, 0.01% SDS, cOmplete EDTA-free Protease Inhibitor Cocktail, 100 mM PMSF) and fragmented to an average length of 500 bp using a Fisher Scientific Model 500 Sonic Dismembrator probe tip sonicator. Soluble chromatin was diluted to 0.2% SDS with IP Dilution Buffer before pre-clearing with 5 µg of mouse IgG conjugated to 20 µl of Dynabeads-Protein G (Thermo Fisher Scientific) followed by immunoprecipitation with 5 µg of anti-FLAG M2 antibody conjugated to 50 µl of Dynabeads-Protein G. Quantitative PCR was carried out on a QuantStudio 7 Flex (Applied Biosystems, Waltham, MA) using locus-specific primers and iTaq SYBR Green Supermix (Bio-Rad Laboratories, Hercules, CA). Primer sequences are listed in *Supplementary file 4*.

## enChIP-MS

The enChIP-MS procedure was performed as described previously (*Fujita and Fujii, 2013*) using chromatin extraction and fragmentation methods from (*Forsberg et al., 2000*) and the following minor modifications. Forty million myoblasts were harvested by trypsinization and lysed in Cell Lysis Buffer (10 mM Tris pH 8.0, 10 mM NaCl, 0.2% IGEPAL-CA630, cOmplete EDTA-free Protease Inhibitor Cocktail, 100 mM PMSF). The isolated nuclei were crosslinked with 1–2% formaldehyde at room temperature for 10–20 min and then lysed in Nuclei Lysis Buffer (50 mM Tris pH 8.0, 10 mM EDTA, 1% SDS, cOmplete EDTA-free Protease Inhibitor Cocktail, 100 mM PMSF). Chromatin was diluted to 0.5% SDS with IP Dilution Buffer and fragmented using a Fisher Scientific Model 500 Sonic Dismembrator probe tip sonicator to an average length of 3 kb. Sonicated chromatin was diluted to 0.2% SDS with IP Dilution Buffer, pre-cleared with 25 µg of mouse IgG conjugated to 100 µl of Dynabeads-Protein G and immunoprecipitated with 70 µg of anti-FLAG M2 antibody conjugated to 180

µl of Dynabeads-Protein G. An additional two Dynabead washes in Low Salt Wash Buffer replaced the high-salt washes. Eluted and precipitated samples were resuspended in SDS sample buffer, boiled and subjected to SDS-PAGE. Entire gel lanes were excised and proteins analyzed using an OrbiTrap Elite mass spectrometer (Thermo Fisher Scientific) coupled to an Easy-nLC II (Thermo Fisher Scientific) at the Fred Hutchinson Cancer Research Center Proteomics Shared Resource. The raw spectra were searched against a UniProt human protein database that also included common contaminants as defined in *Mellacheruvu et al. (2013)* using Proteome Discoverer 1.4 software (Thermo Fisher Scientific) to generate peptide-spectrum matches. The number of peptides that mapped to each protein was summarized to generate a 'pseudoquant' metric. Proteins with at least one peptide-spectrum match in two experimental replicates were carried forward for further analysis, after filtering out common contaminants. Finally, the UniProt annotations for Function and Sub-cellular location were used to restrict the analysis to only the nuclear proteins to enrich for biologically relevant, nuclear interactions. The R code used for the proteomics data analysis can be accessed via github (https://github.com/sjaganna/2017-campbell_et_al) (*Jagannathan, 2017*). The gRNA sequences are listed in *Supplementary file 4*.

## GO category analysis

GO analysis was carried out with the PANTHER classification system (*Mi et al., 2016*) using the statistical overrepresentation test against all human genes and the complete GO Biological process annotation. p-Values were corrected for multiple hypothesis testing using the Bonferroni correction.

## ChIP-qPCR

The occupancy of NuRD complex components and acetyl-Histone H4 was determined using cross-linked ChIP coupled with micrococcal nuclease digestion as described previously (*Skene and Henikoff, 2015*). For acetyl-Histone H4 samples, the Lysis Buffer and IP Dilution Buffer were supplemented with 10 mM sodium butyrate. Quantitative PCR was carried out on a QuantStudio 7 Flex using locus-specific primers and iTaq SYBR Green Supermix. Primer sequences are listed in *Supplementary file 4*.

## siRNA transfections

Flexitube and ON-TARGETplus duplex siRNAs were obtained from Qiagen (Hilden, Germany) or GE Dharmacon (Lafayette, CO), respectively. Transfections of siRNAs into myoblasts and iPS cells were carried out using Lipofectamine RNAiMAX (Invitrogen) according to the manufacturer's instructions. A double transfection protocol was followed in myoblasts to ensure efficient depletion of pre-existing proteins. Briefly, cells were seeded at ~30% confluence in six-well plates and transfected ~20 hr later with 6 µl Lipofectamine RNAiMAX and 25 pmol of either gene-specific siRNA(s) or a scrambled non-silencing control siRNA diluted in 125 µl Opti-MEM Reduced Serum Medium (Thermo Fisher Scientific). Forty-eight hours following this, myoblasts were transfected a second time and harvested for RNA analysis 48–72 hr later. In iPS cells, the same procedure was followed except cells were treated with 10 µM Y-27632 ROCK inhibitor (Miltenyi Biotec, Auburn, CA) for 24 hr before being trypsinized and seeded in mTeSR1 medium (STEMCELL Technologies, Vancouver, BC) at $1 \times 10^5$ cells/well on Matrigel (Corning Life Science, Tewksbury, MA)-coated six-well plates, and were harvested 48 hr after a single transfection. The sequences of siRNAs are listed in *Supplementary file 4*.

## RNA isolation and RT-qPCR

Total RNA was extracted from whole cells using the RNeasy Mini Kit (Qiagen) according to the manufacturer's instructions. The isolated RNA was treated with DNase I (Thermo Fisher Scientific), heat inactivated, and reverse transcribed into cDNA using Superscript III (Thermo Fisher Scientific) and oligo(dT) primers (Invitrogen) following the manufacturer's protocol. Quantitative PCR was carried out on a QuantStudio 7 Flex using primers specific for each mRNA and iTaq SYBR Green Supermix. The relative expression levels of target genes were normalized to that of the reference genes *RPL27*, *RPL13A* or *GAPDH* by using the delta-delta-Ct method (*Livak and Schmittgen, 2001*) after confirming equivalent amplification efficiencies of reference and target molecules. Primer sequences are listed in *Supplementary file 4*.

## Antibodies

The following antibodies were used: α-Tubulin (T9026); Acetyl-Histone H4 (06–866 lot#2554112, EMD Millipore (Billerica, MA)); CHD4 (A301-081A, Bethyl Laboratories (Montgomery, TX)); FITC anti-mouse (715-095-150 lot#115855, Jackson ImmunoResearch (West Grove, PA)); FLAG M2 (F1804 lot#SLBG5673V and lot#124K6106); FLAG M2 (F3165 lot#SLBL1237V); HDAC2 (ab7029, lot#GR88809-7, Abcam (Cambridge, UK)); HRP anti-mouse (115-035-146, Jackson ImmunoResearch); MBD2 (A301-632A, Bethyl); mouse IgG (315-005-003 lot#120058, Jackson ImmunoResearch); MTA2 (ab8106 lot#GR185489-3, Abcam); TRITC anti-rabbit (711-025-152 lot#114768, Jackson ImmunoResearch); rabbit monoclonal antibodies against DUX4 (E5-5 and E14-3) were produced in collaboration with Epitomics and are described elsewhere (*Geng et al., 2012*).

## Statistical analysis

All collected data were included in the analyses. Statistical significance was determined using Mann-Whitney *U* or Wilcoxon signed-rank tests, as indicated in the corresponding figure legends. As is convention, at least three biological replicates per condition were used for ChIP-qPCR and RT-qPCR, as indicated. Here a biological replicate is defined as an independent culture of cells that was separately manipulated and subsequently analyzed. The enChIP-MS studies were multiple singleton experiments performed using several different gRNA that targeted the same genomic locus, as described. No statistical methods were used to predetermine sample size. Masking was not used during group allocation, data collection or data analysis.

## Accession codes

The mass spectrometry proteomics data have been deposited to the ProteomeXchange Consortium (http://proteomecentral.proteomexchange.org) via the PRIDE partner repository (*Vizcaíno et al., 2016*) with the dataset identifier PXD006839.

## Acknowledgements

The authors thank Phil Gafken and Yuko Ogata of the Fred Hutchinson Cancer Research Center Proteomics Shared Resource for their assistance in planning and executing the enChIP-MS. We would also like to acknowledge the members of the Tapscott lab for helpful discussion throughout the project.

## Additional information

### Funding

| Funder | Grant reference number | Author |
|---|---|---|
| National Cancer Institute | T32CA009657 | Amy E Campbell |
| National Human Genome Research Institute | T32HG00035 | Sean C Shadle |
| National Institute of General Medical Sciences | T32GM007270 | Sean C Shadle |
| FSH Society | FSHS-22014-01 | Sujatha Jagannathan |
| National Human Genome Research Institute | T32HG000035 | Rebecca Resnick |
| Eunice Kennedy Shriver National Institute of Child Health and Human Development | T32HD007183 | Rebecca Resnick |
| National Institute of Neurological Disorders and Stroke | P01NS069539 | Rabi Tawil<br>Silvère M van der Maarel<br>Stephen J Tapscott |
| Prinses Beatrix Spierfonds | W.OP14-01 | Silvère M van der Maarel |
| Spieren voor Spieren | | Silvère M van der Maarel |

| National Institute of Arthritis and Musculoskeletal and Skin Diseases | R01AR066248 | Silvère M van der Maarel Stephen J Tapscott |
| National Institute of Arthritis and Musculoskeletal and Skin Diseases | R01AR045203 | Stephen J Tapscott |
| Friends of FSH Research | | Stephen J Tapscott |

The funders had no role in study design, data collection and interpretation, or the decision to submit the work for publication.

### Author contributions
Amy E Campbell, Conceptualization, Data curation, Formal analysis, Funding acquisition, Validation, Investigation, Visualization, Methodology, Writing—original draft, Project administration, Writing—review and editing; Sean C Shadle, Conceptualization, Funding acquisition, Validation, Investigation, Methodology, Writing—review and editing; Sujatha Jagannathan, Resources, Data curation, Software, Formal analysis, Funding acquisition, Writing—review and editing; Jong-Won Lim, Resources, Methodology, Writing—review and editing; Rebecca Resnick, Funding acquisition, Investigation, Writing—review and editing; Rabi Tawil, Silvère M van der Maarel, Resources, Funding acquisition, Writing—review and editing; Stephen J Tapscott, Conceptualization, Supervision, Funding acquisition, Writing—original draft, Project administration, Writing—review and editing

### Author ORCIDs
Amy E Campbell (iD) https://orcid.org/0000-0001-6513-5836
Sujatha Jagannathan (iD) http://orcid.org/0000-0001-9039-2631
Rebecca Resnick (iD) http://orcid.org/0000-0001-5804-4418
Stephen J Tapscott (iD) http://orcid.org/0000-0002-0319-0968

### Decision letter and Author response
Decision letter https://doi.org/10.7554/eLife.31023.047
Author response https://doi.org/10.7554/eLife.31023.048

# Additional files

### Supplementary files
• Supplementary file 1. Proteins identified by enChIP-MS. The table lists the gene name, corresponding number of peptides recovered (pseudoquant), and percent coverage for each protein identified by enChIP-MS of nine independent FLAG-dCas9 immunoprecipitations from various gD4Z4- or gMYOD1-expressing myoblast cell lines.
DOI: https://doi.org/10.7554/eLife.31023.039

• Supplementary file 2. Gene ontology analysis of D4Z4-associated proteins. The table shows the gene ontology (GO) biological process categories enriched among the D4Z4-associated proteins identified by enChIP-MS, along with the number of observed versus expected proteins in each category and the associated fold enrichment score and *p*-value.
DOI: https://doi.org/10.7554/eLife.31023.040

• Supplementary file 3. Characteristics of myoblast cell lines used in this study. The table summarizes details of the muscle cell lines used for this study.
DOI: https://doi.org/10.7554/eLife.31023.041

• Supplementary file 4. Oligonucleotide sequences. The table lists all oligonucleotides used in this study, including gRNAs, siRNAs, shRNAs and primers used for enChIP-qPCR, ChIP-qPCR and RT-qPCR.
DOI: https://doi.org/10.7554/eLife.31023.042

• Transparent reporting form
DOI: https://doi.org/10.7554/eLife.31023.043

## Major datasets

The following dataset was generated:

| Author(s) | Year | Dataset title | Dataset URL | Database, license, and accessibility information |
|---|---|---|---|---|
| Campbell AE, Jagannathan S, Tapscott SJ | 2017 | NuRD and CAF-1-mediated silencing of the D4Z4 array is modulated by DUX4-induced MBD3L proteins | http://proteomecentral. proteomexchange.org/ cgi/GetDataset?ID= PXD006839 | Publicly available at ProteomeXchange (accession no. PXD006 839) |

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
