## [Decision Letter]

Thank you for submitting your article "MBD3L2 relief of NuRD and CAF-1 silencing of the D4Z4 array amplifies DUX4 expression" for consideration by *eLife*. Your article has been reviewed by three peer reviewers, and the evaluation has been overseen by a Reviewing Editor and Jessica Tyler as the Senior Editor. The reviewers have opted to remain anonymous.

The reviewers have discussed the reviews with one another and the Reviewing Editor has drafted this decision to help you prepare a revised submission.

The macrosatellite repeat locus D4Z4 harbors the retrogene encoding the DUX4 transcription factor. The expression of DUX4 in early embryos activates a cleavage-specific transcriptional program. DUX4 is normally repressed in somatic tissues but D4Z4 repeat contraction or mutations in chromatin regulators of D4Z4 result in facioscapulohumeral muscular dystrophy (FSHD) due to mis-expression of DUX4 in skeletal muscle. In this manuscript the authors have applied a powerful proteomics approach using LC-MS to identify the proteins that are bound to the D4Z4 macrosatellite locus in myogenic cells. They validate these factors and explore their regulatory potential through siRNA knockdowns. In this way they identify components of the NURD complex and of the chromatin assembly factor (CAF-1), bound to the D4Z4 locus. Down-regulation of HDAC1 and HDAC2 induces Dux4 and ZSCAN4 expression, similarly to CHD4. Knock down of CAF-1 p150 or p60 subunits induces Dux4 and ZSCAN4 expression in FSHD mutant myoblasts. In the last figure, they show that siRNA for Mbd2, Chd4, CAF-1 subunits and HDAC1 and 2, induce DUX4 expression in human iPS cells. The authors go on to propose a new mechanism, via MBD3L2 induction and competition, for amplification and spread of the DUX4 signal after an initial myonuclear burst.

Overall this study is impressive both technically, for the approach the authors developed to identify proteins at the D4Z4 locus and for the therapeutic perspectives as the proteins discovered could provide future therapeutic targets for FSHD. The mechanistic insights as to how the factors identified influence D4Z4 are still quite limited. Nevertheless, this manuscript presents interesting new data and opens up the field for FSHD research, and is also of potential interest for macrosatellite control more generally.

There are however a number of issues that need to be addressed, listed below. In particular, the number of cell lines tested is rather limited – the claims would be more convincing if further FSHD1 and FSHD2 lines were tested. Also enthusiasm was limited for the MBD3L2 feed-forward mechanism and reviewers felt that it occupied too central of a position in the title and Abstract. Focusing on the MBD3L2 mechanism detracts from the impact of the impressive work of identifying the D4Z4 proteome, testing its components, and showing that NURD and CAF-1 are essential for repression of the locus, even in FSHD cells. Indeed as the manuscript stands, the title, impact statement and Abstract overstate the importance of MBD3L: "The early embryonic transcription factor DUX4 induces the expression of MBD3L genes that inhibit the NuRD complex revealing a mechanism of amplifying DUX4 expression in facioscapulohumeral dystrophy". The role of MBD3L would need to be developed further in order to support such claims, as the size of the effects seen was not major. At the very least, the title, Abstract and impact text need to be modified, and should better reflect the main topic of this paper, which is the identification of factors associated with D4Z4.

Essential revisions:

1) D4Z4 underlies FSHD and the authors explore disease-relevant cell lines FSHD1 and FSHD2. In FSHD, it is common to see variability from patient to patient, not just clinically but also in patient-derived cells. Most of the analysis is performed on a single FSHD and a single control myoblast line. By using single FSHD1 and FSHD2 lines, it does not seem that the authors can distinguish between individual patient differences or differences specific to the disease lesions in FSHD1 and FSHD2 (e.g., in the MBD1 and MBD2 experiments). If possible, more individual cell lines (both FSHD1 and FSHD2) should be used for the validations and in order to support the claims about MBD3L2 more convincingly in particular.

2) For the knockdown analyses, the authors are commended for providing comprehensive control data for all of their knockdowns in each experiment. However the study uses n=3 throughout, and performs t-tests to demonstrate significance. As sample size decreases, the t-test becomes extremely sensitive to the assumption of normality, and although a sample of 3 can be consistent with a normal distribution of the population, it cannot provide high confidence that the population is in fact normally distributed. Therefore, the authors should either increase sample size, or use a nonparametric test to calculate significance.

3) Some of the knockdowns were much less effective compared to others, for example CHD3, MBD3, SETDB1 – it seems that this establishes caveats that should be mentioned by the authors.

4) Could the authors please provide the provenance of the eMHF2 iPS line? If this is the control cell line – please make this point clearly. Also, please address the inconsistency between the statement "we have previously shown that DUX4 is expressed in iPS cells" and the data that it is actually quite repressed in iPS cells until NuRD and CAF-1 are disabled.

5) Similar to point 2, above, the effect of NuRD and CAF-1 inhibition should be evaluated in at least two additional iPS cell lines.

6) In the mass spec summaries (Table 1 and Supplementary file 1), the authors indicate the number of peptides detected for each protein. This metric biases in favor of larger proteins, as they break into more peptides. Could the authors please add% coverage as an additional column alongside number of reads?

7) The authors should improve their interpretation (and discussion) of the function of CAF-1 and Nurd. CAF-1 regulates the early embryo cleavage expression programme, including ZSCAN4 and Dux, in mouse ES cells (Hendrickson et al., 2017 and Ishiuchi et al., 2015). Likewise, Trim28, Kdm1a/Lsd1 and HDAC activity (TSA, VPA) have been shown to regulate ZSCAN4 and the 'early cleavage programme' in the mouse (Macfarlan et al. Nature 2014). Neither the Macfarlan, nor the Ishiuchi references are included in the text – for example in the discussion on CAF-1 (Discussion). Furthermore, there are conceptual mistakes in the current text concerning CAF1, concerning its known biochemical and cellular activities (see reviewer 3's comments for more detailed information).

8) Chromatin remodelers and other positive transcriptional regulators might be expected to be present at the contracted D4Z4 array in FSHD cells. The association with repressed D4Z4s in healthy cells is therefore surprising. It would be interesting if the authors commented on the enrichment of SMARCA5, BRD3, and BRD4 at the D4Z4 arrays in healthy myoblasts.

9) There are several overstatements and over interpretations in this paper, particularly the role of MBDL3 as mentioned above and also:

- The authors cannot claim to have "uncovered a mechanism for relieving D4Z4 epigenetic repression in the early embryo" – please change.

- Interpretation in the last paragraph of the subsection “Components shared by the NuRD and CAF-1 complexes mediate D4Z4 repeat repression”, on CAF-1 involved in multiple transcriptionally inhibitory factors needs revision.

- Subsection “Proteins that repress the D4Z4 array in myoblasts also silence DUX4 in iPS cells”. There is no evidence in Hendrickson and Whiddon references that "DUX4 […] activates a cleavage-stage specific transcriptional program in human 4-cell embryos". Please rephrase.

- The conclusion “These results indicate that NuRD and CAF-1 complexes that silence D4Z4 array in muscle cells also contribute to the regulation of this locus during early development” is not founded. Please remove.

10) The impact and value of the study is not conveyed by the title neither by the impact statement and Abstract. The first 5 figures have nothing to do with MBD3L2 relieving silencing. This notion comes up at the end, in Figure 6, and is not developed, certainly not to the point of focusing the title on it. In view of the importance of the purification method to the success of the study, this should be better highlighted in the title and Abstract.

11) Similarly, the schematic describing the purification could be included in Figure 1 proper, rather than in the supplemental data for Figure 1.

*Reviewer #1:*

This work has two novel and significant components: 1) Using a clever strategy for unbiased biochemical enrichment, the authors identified, and then verified, many of the factors bound at the FSHD-associated D4Z4 macrosatellite in myogenic cells in vivo; 2) identification of a new mechanism, via MBD3L2 induction and competition, for amplification and spread of the DUX4 signal after an initial myonuclear burst. This study represents overall solid work with minor concerns. In FSHD, it is common to see variability from patient to patient, not just clinically but also in patient-derived cells. By using single FSHD1 and FSHD2 lines, it does not seem that the authors can distinguish between individual patient differences or differences specific to the disease lesions in FSHD1 and FSHD2 (e.g., in the MBD1 and MBD2 experiments). Similarly, the crux of the paper and rationale for the title is Figure 6, and the key experiment is done only in FSHD2 and not in FSHD1 cells. The level of knockdown is great, but the effect on DUX4-fl expression is slight (~25%) in what is basically the most relevant therapeutic target cell (myoblasts); is this because they are using FSHD2 cells? For potential therapeutic targeting, FSHD1 cells would seem to be the better choice (and possibly the cells in which the endogenous mechanism is most important). Alternatively, it seems worth looking at DUX4-fl transcripts at a later timepoint (perhaps 48 hours is too early to see much of an effect). Overall, the data does support the claims, which are significant, and the biochemical screening stands on its own. Some of the differences found in validation and the MBD3L claims would be more convincing if performed in more individual cell lines (both FSHD1 and FSHD2), yet that likely would not change the big picture end results, just some of the details.

*Reviewer #2:*

Enthusiasm is very high for this paper, which identifies proteins bound to D4Z4 and evaluates their regulatory potential through knockdowns. The discovery of the important role of the CAF/1 and NuRD repressive complexes will be of great interest to the FSHD field.

There are a few significant concerns, however:

1) The impact and value of the study is not conveyed by the title, and to some degree, neither by the Abstract. The first 5 figures have nothing to do with MBD3L2 relieving silencing. This notion comes up at the end, in Figure 6, and is really not very developed, certainly not to the point of focusing the title on it. The beautiful magnum opus of purifying D4Z4 chromatin, identifying and then testing a large number of factors is not even mentioned in the Abstract. The long first half of the Abstract does not actually address the contents of the manuscript, but rather serves as an introduction – it could be condensed by half.

2) In view of the importance of the purification method to the success of the study, I suggest including the schematic describing the purification in Figure 1 proper, rather than in the supplemental data for Figure 1.

3) Rigor and reproducibility. First, the authors are commended for providing comprehensive control data for all of their knockdowns in each experiment. Nevertheless, there is a concern with rigor. In the current environment, it is becoming more difficult to justify extremely low sample size experiments. The study uses n=3 throughout, and performs t-tests to demonstrate significance. As sample size decreases, the t-test becomes extremely sensitive to the assumption of normality, and although a sample of 3 can be consistent with a normal distribution of the population, it cannot provide high confidence that the population is in fact normally distributed. Therefore, the authors should either increase sample size, or use a nonparametric test to calculate significance.

3) Some of the knockdowns were much less effective compared to others, for example CHD3, MBD3, SETDB1 – it seems that this establishes caveats that should be mentioned by the authors.

4) Most of the analysis is performed on a single FSHD and a single control myoblast line. Could the authors please take whatever in their view is the most significant knockdown (or combination) culminating from this study and demonstrate its effectiveness in a small panel of FSHD and control cell lines? This will give confidence that the results will be broadly relevant.

5) Could the authors please provide the provenance of the eMHF2 iPS line. I assumed that this is a control cell line – please make this point clearly. Also, please address the inconsistency between the statement "we have previously shown that DUX4 is expressed in iPS cells" and the data that it is actually quite repressed in iPS cells until NuRD and CAF/1 are disabled.

6) Similar to point 4, above, please evaluate the effect of NuRD and CAF/1 inhibition in at least two additional iPS or ES cell lines.

7) In the mass spec summaries (Table 1 and Supplementary file 1), the authors indicate the number of peptides detected for each protein. This metric biases in favor of larger proteins, as they break into more peptides. Could the authors please add% coverage as an additional column alongside number of reads?

*Reviewer #3:*

The authors report a locus proteomics approach using LC-MS that identifies components of the NURD complex and of the chromatin assembly factor (CAF1), bound to the D4Z4 locus in myoblasts. They validate their targets by performing siRNA for several subunits of NURD, alone or in combination. Based on this, they conclude that combined down-regulation of HDAC1 and HDAC2 induces Dux4 (and ZSCAN4) expression, similarly to CHD4. Likewise, they show that siRNA for p150 or p60 subunits of CAF-1 induces Dux4 and ZSCAN4 expression in FSHD mutant myoblasts, as measured by RT-QPCR. Finally, in the last figure, they show that siRNA for Mbd2, Chd4, CAF-1 subunits and HDAC1 and 2, induce DUX4 expression in human iPS cells.

The work is properly controlled, and the method by itself is very interesting. While I do not have specific technical comments on the data, without any further characterisation (for example, biochemical) on how Nurd or CAF-1 act to regulate D4Z4, I find the manuscript not very novel and a little preliminary (in short, MassSpec of the D4Z4 locus and many siRNA coupled to RT-QPCR of the subunits of two complexes to validate their MassSPec). The question that remains is how does any of these complexes actually regulate D4Z4 chromatin structure and DUX4 expression?

CAF-1 has already been shown to regulate the 'early cleavage expression' programme, including ZSCAN 4 and Dux, in mouse ES cells (Hendrickson et al., 2017 and Ishiuchi et al., 2015). Likewise, Trim28, Kdm1a/Lsd1 and HDAC activity (TSA, VPA) have been shown to regulate ZSCAN4 and the 'early cleavage programme' in the mouse (Macfarlan et al. Nature 2014). Neither the Macfarlan, nor the Ishiuchi references are included in the text. I find this particularly surprising, especially considering the discussion on CAF1 (Discussion).

At the very least, the authors should improve their interpretation (and discussion) of the function of CAF-1 and Nurd, the former in which I find important conceptual mistakes, based on the known biochemical and cellular activities of CAF-1. I also find several of their conclusions/interpretations overstated, which will unnecessarily confuse the readers, and which the authors should correct, as per my comments below.

CAF-1 is not a transcriptionally repressive complex (subsection “Silencing the D4Z4 array requires components of the MBD1/CAF-1 complex”, first paragraph; subsection “Silencing the D4Z4 array requires components of the MBD1/CAF-1 complex”, last paragraph; subsection “Components shared by the NuRD and CAF-1 complexes mediate D4Z4 repeat repression”, first paragraph), but a chromatin assembly complex which functions only in parallel to DNA synthesis of the whole genome, mainly in S-phase, and under some circumstances (e.g. DNA repair), during G1. In fact, its expression is regulated throughout the cell cycle (see works by Paul Kauffman, Torsten Krude and Genevieve Almouzni). Along these lines, the conceptual interpretation of CAF-1 function is misleading. Also, the authors do not really show how CAF-1 (or NuRD) regulate chromatin function: for example, do myoblasts proliferate, and hence CAF1 function may be related to its chromatin assembly activity?

I find the following sentences overstated, and there is no data in the manuscript that supports such conclusions, please rephrase/remove:

1) The authors have not "uncovered a mechanism for relieving D4Z4 'epigenetic repression in the early embryo".

2) Interpretation in the last paragraph of the subsection “Components shared by the NuRD and CAF-1 complexes mediate D4Z4 repeat repression”, on CAF-1 involved in multiple transcriptionally inhibitory factors needs revision.

3) Subsection “Proteins that repress the D4Z4 array in myoblasts also silence DUX4 in iPS cells”. There is no evidence in Hendrickson and Whiddon references that "DUX4 […] activates a cleavage-stage specific transcriptional program in human 4-cell embryos". Please rephrase.

4) The conclusion “These results indicate that NuRD and CAF-1 complexes that silence D4Z4 array in muscle cells also contribute to the regulation of this locus during early development” is not founded. Please remove.

---

## [Author Response]

Essential revisions:1) D4Z4 underlies FSHD and the authors explore disease-relevant cell lines FSHD1 and FSHD2. In FSHD, it is common to see variability from patient to patient, not just clinically but also in patient-derived cells. Most of the analysis is performed on a single FSHD and a single control myoblast line. By using single FSHD1 and FSHD2 lines, it does not seem that the authors can distinguish between individual patient differences or differences specific to the disease lesions in FSHD1 and FSHD2 (e.g., in the MBD1 and MBD2 experiments). If possible, more individual cell lines (both FSHD1 and FSHD2) should be used for the validations and in order to support the claims about MBD3L2 more convincingly in particular.

We appreciate the reviewers’ caution and suggestions. In our original manuscript we tried to be careful not to generalize the possible FSHD context-dependent difference we observed upon MBD1 and MBD2 knockdown in single FSHD1 and FSHD2 cell lines. In this revision, we have added four figure supplements (Figure 3—figure supplement 4–Figure 3—figure supplement 7) showing siRNA depletion of CHD4, MBD2, CHAF1A, and MBD1 in one additional FSHD1 and four additional FSHD2 cell lines. These new studies reveal that knockdown of MBD2 de-represses DUX4 in all of these cell lines, whereas knockdown of MBD1 de-repressed DUX4 in two of the new FSHD2 cell lines and in the new FSHD1 cell line, as noted in the revised Results. The added text reads:

“To extend these studies, we depleted CHD4, CHAF1A, MBD2, or MBD1 in five additional FSHD cell lines: one FSHD1 cell line (54-2) with three 4qA D4Z4 repeats (compared to the 8 repeats of the MB073 line), and four FSHD2 lines (2305, 2453, 2338, and 1881) with different *SMCHD1* mutations and repeat sizes ranging from 11-15 D4Z4 units (Supplementary file 3). […] Taken together, these data indicate the combined roles of the NuRD and CAF-1 complexes in repressing DUX4, and that the relative necessity of specific components of each pathway might vary depending on the cellular context, or possibly the efficiency of each knockdown.” Figure legends have been updated accordingly, as has the Discussion (first paragraph and Essential revision 3, below).

Also in our original manuscript we demonstrated that forced expression of MBD3L2 de-repressed DUX4 in both MB073 FSHD1 and MB200 FSHD2 myoblasts; however, we only used the MB200 FSHD2 cells to test whether knockdown of MBD3L genes suppressed DUX4 expression in muscle cells. In this revision, we confirm that MBD3L depletion in the MB073 FSHD1 cell line resulted in a ~50% reduction in DUX4 expression. These new data are shown in new Figure 6 and Figure 6—figure supplement 1.

Additionally, in the original manuscript we presented combined data from three independent experiments using control or MBD3L shRNA-expressing MB200 FSHD2 muscle cell lines, whereas in this revision we display three independent datasets for both the FSHD1 and FSHD2 cell lines as new Figure 6, Figure 6—figure supplement 1 and Figure 6—figure supplement 2. We feel that this better communicates the robustness of the observed effect of MBD3L gene knockdown on DUX4 levels. The Results have been updated and the text now reads:

“When cultured in low mitogen differentiation media, myoblasts fuse to form multinucleated myotubes, and DUX4 expression increases in FSHD myotubes compared to myoblasts (Balog et al., 2015). […] Together, these data implicate MBD3L2 in the regulation of the D4Z4 array and demonstrate that endogenous DUX4-induced MBD3L proteins contribute to the amplification of DUX4 expression in FSHD myotubes.”

Figure legends have been revised accordingly.

2) For the knockdown analyses, the authors are commended for providing comprehensive control data for all of their knockdowns in each experiment. However the study uses n=3 throughout, and performs t-tests to demonstrate significance. As sample size decreases, the t-test becomes extremely sensitive to the assumption of normality, and although a sample of 3 can be consistent with a normal distribution of the population, it cannot provide high confidence that the population is in fact normally distributed. Therefore, the authors should either increase sample size, or use a nonparametric test to calculate significance.

We thank the reviewers for bringing to our attention the fact that the normality assumption should not be ignored when dealing with small sample sizes such as the 3 replicates per group used throughout our study. We have therefore revised the manuscript to use a Mann-Whitney *U* test (the nonparametric counterpart to the two-sample *t*-test) or Wilcoxon signed-rank test (the nonparametric counterpart to the one-sample *t*-test) for the significance calculations shown in Figure 1, Figure 2, Figure 4, Figure 5, Figure 6, Figure 2—figure supplement 2, Figure 2—figure supplement 5 and Figure 3—figure supplement 4–Figure 3—figure supplement 7. The Materials and methods and figure legends have been updated accordingly (see subsection “Statistical analysis”).

3) Some of the knockdowns were much less effective compared to others, for example CHD3, MBD3, SETDB1 – it seems that this establishes caveats that should be mentioned by the authors.

The reviewers raise an excellent point relevant for all knockdown studies, and we address this in the revised Results (see subsection “Silencing the D4Z4 array requires components of the MBD1/CAF-1 complex”, last paragraph and Essential revision 1, above) and Discussion. The Discussion text now reads:

“To some extent, each complex appears to have a parallel, or redundant, function in DUX4 repression because knockdown of both pathways was necessary to induce DUX4 expression in MB2401 control myoblasts. […] However, the variable efficiencies of the individual knockdowns in each cell type and experiment might also contribute to these apparent differences.”

4) Could the authors please provide the provenance of the eMHF2 iPS line? If this is the control cell line – please make this point clearly. Also, please address the inconsistency between the statement "we have previously shown that DUX4 is expressed in iPS cells" and the data that it is actually quite repressed in iPS cells until NuRD and CAF-1 are disabled.

We have clarified in the Materials and methods that the eMHF2 iPS cell line was obtained from the University of Washington Institute for Stem Cell and Regenerative Medicine Tom and Sue Ellison Stem Cell Core as a control (non-FSHD) iPS cell line (see subsection “Cell culture and reagents”). In the Results, we also now state that the eMHF2 iPS cell line was derived from an unaffected individual (see subsection “Proteins that repress the D4Z4 array in myoblasts also silence DUX4 in iPS cells”, first paragraph). Additionally, the Results have been revised to better elucidate for the reader what is known about DUX4 expression in early developmental contexts. The revised Results now read:

“We previously reported that DUX4 is expressed at very low levels in human iPS cell populations (Snider et al., 2010) and, similar to the expression pattern in FSHD myoblasts, this represents the occasional expression in a small number of cells (JWL, unpublished data). […] To determine whether factors responsible for silencing the D4Z4 repeat in myoblasts have a similar function in a model of early development, we knocked down components of the NuRD and CAF-1 complexes in human eMHF2 iPS cells, which were derived from an unaffected (non-FSHD) individual, and assessed the impact on DUX4 expression.”

5) Similar to point 2, above, the effect of NuRD and CAF-1 inhibition should be evaluated in at least two additional iPS cell lines.

We have added two figure supplements (Figure 5—figure supplement 2–Figure 5—figure supplement 3) showing siRNA depletion of NuRD and CAF-1 complex components in a human fibroblast cell line and three iPS cell lines derived from these fibroblasts. These new studies confirm that knockdown of CHD4 or CHAF1A activate DUX4 in the iPS cell lines, as noted in the revised Results. The text now reads:

“To determine whether iPS cells have a greater necessity for NuRD and CAF-1 components to maintain DUX4 repression compared to somatic cells, we transduced a human foreskin fibroblast cell line (HFF3) with the reprogramming factors Oct4, *Sox2*, Nanog, and Lin28 to generate isogenic iPS cell clones (Figure 5—figure supplement 2). […] These results indicate that the NuRD and CAF-1 complexes that silence the D4Z4 macrosatellite array in muscle cells also contribute to the regulation of this locus in human iPS cells, and that iPS cells have decreased D4Z4 repression compared to their somatic counterpart, similar to the decreased repression in FSHD myoblasts compared to control myoblasts.” The Materials and methods and figure legends have been updated accordingly (see subsection “Cell culture and reagents”).

6) In the mass spec summaries (Table 1 and Supplementary file 1), the authors indicate the number of peptides detected for each protein. This metric biases in favor of larger proteins, as they break into more peptides. Could the authors please add% coverage as an additional column alongside number of reads?

We have updated the mass spectrometry summaries in Table 1 and Supplementary file 1 to include percent coverage information.

7) The authors should improve their interpretation (and discussion) of the function of CAF-1 and Nurd. CAF-1 regulates the early embryo cleavage expression programme, including ZSCAN 4 and Dux, in mouse ES cells (Hendrickson et al., 2017 and Ishiuchi et al., 2015). Likewise, Trim28, Kdm1a/Lsd1 and HDAC activity (TSA, VPA) have been shown to regulate ZSCAN4 and the 'early cleavage programme' in the mouse (Macfarlan et al. Nature 2014). Neither the Macfarlan, nor the Ishiuchi references are included in the text – for example in the discussion on CAF-1 (Discussion). Furthermore, there are conceptual mistakes in the current text concerning CAF1, concerning its known biochemical and cellular activities (see reviewer 3's comments for more detailed information).

We have revised the Discussion to better reflect the literature and points made by the reviewers regarding mouse ES cells. This paragraph in the Discussion now reads:

“Although little is known about the regulation of DUX4 expression in cleavage-stage embryos and the testis luminal cells, it is evident from this study that the expression of DUX4 in a small percentage of iPS cells or ES cells shares mechanisms of molecular regulation with skeletal muscle cells. […] The fact that inhibiting NuRD or CAF-1 activity potentiates stem cell reprogramming in mouse ES/iPS cells and, as shown in this report, potentiates human DUX4 expression, suggests that DUX4 itself might facilitate reprogramming to the naïve state and that mouse Dux and human DUX4 might be subject to similar regulation, a finding note entirely obvious given that these retrogenes are thought to have been generated by independent retrotranspositions of the parental DUXC gene, as noted above.”

Regarding the conceptual mistakes concerning CAF-1, we agree that the original submission had a confusing presentation of the mechanistic role of the CAF-1 complex. Throughout the revised text we have tried to more clearly state its role as a chromatin assembly complex and that the knockdown studies were performed in myoblasts. In addition, in the first paragraph of the Discussion we have added a sentence reading:

“It is also important to note that CAF-1 is a chromatin assembly complex and that knockdowns were performed in replicating myoblasts; therefore, CAF-1 knockdown might not have the same consequence in post-mitotic myotubes.”

8) Chromatin remodelers and other positive transcriptional regulators might be expected to be present at the contracted D4Z4 array in FSHD cells. The association with repressed D4Z4s in healthy cells is therefore surprising. It would be interesting if the authors commented on the enrichment of SMARCA5, BRD3, and BRD4 at the D4Z4 arrays in healthy myoblasts.

We have added text to the Discussion, which reads:

“The presence of chromatin remodelers and positive transcriptional regulators, such as SMARCA5, BRD3 and BRD4, at the D4Z4 locus in the control cells used for the enChIP also indicates a dynamic balance between activators and repressors, which is consistent with the identification of sense and anti-sense transcripts associated with the D4Z4 repeats in both control and FSHD cells (Snider et al., 2010).”

9) There are several overstatements and over interpretations in this paper, particularly the role of MBDL3 as mentioned above and also:- The authors cannot claim to have "uncovered a mechanism for relieving D4Z4 epigenetic repression in the early embryo" – please change.

This wording has been deleted.

- Interpretation in the last paragraph of the subsection “Components shared by the NuRD and CAF-1 complexes mediate D4Z4 repeat repression”, on CAF-1 involved in multiple transcriptionally inhibitory factors needs revision.

This has been revised to read: “Taken together, these data indicate that D4Z4 array silencing is mediated by multiple chromatin regulatory factors that act together with core components of the NuRD complex and also depend on the CAF-1 chromatin assembly complex to achieve full epigenetic repression.”

- Subsection “Proteins that repress the D4Z4 array in myoblasts also silence DUX4 in iPS cells”. There is no evidence in Hendrickson and Whiddon references that "DUX4 […] activates a cleavage-stage specific transcriptional program in human 4-cell embryos". Please rephrase.

We have reworded this sentence to read: “We have more recently shown that DUX4 is present in 4-cell human embryos and that when expressed in iPS cells or muscle cells it activates a cleavage-stage transcriptional program similar to the program expressed in a subset of ‘naïve’ iPS or embryonic stem (ES) cells (Hendrickson et al., 2017; Whiddon et al., 2017).”

- The conclusion “These results indicate that NuRD and CAF-1 complexes that silence D4Z4 array in muscle cells also contribute to the regulation of this locus during early development” is not founded. Please remove.

We have changed the sentence to more accurately state: “These results indicate that the NuRD and CAF-1 complexes that silence the D4Z4 macrosatellite array in muscle cells also contribute to the regulation of this locus in human iPS cells, and that iPS cells have decreased D4Z4 repression compared to their somatic counterpart, similar to the decreased repression in FSHD myoblasts compared to control myoblasts.”

10) The impact and value of the study is not conveyed by the title neither by the impact statement and Abstract. The first 5 figures have nothing to do with MBD3L2 relieving silencing. This notion comes up at the end, in Figure 6, and is not developed, certainly not to the point of focusing the title on it. In view of the importance of the purification method to the success of the study, this should be better highlighted in the title and Abstract.

The focus of the original title was the result of a hurried attempt to shorten the title to fit the character limit encountered when trying to submit the original manuscript. We agree with the criticism of the reviewers, however, this character limit makes it difficult to convey the full results of the study with relative weighting. In this revision, we originally retitled the manuscript as “NuRD and CAF-1 mediated silencing of the D4Z4 array is modulated by DUX4-induced MBD3L proteins”; however, *eLife* Editorial Support would not accept this title because it used unfamiliar abbreviations or acronyms. Therefore, we further changed the title to read “CHD4 and CHAF1A/B mediated silencing of the D4Z4 array is modulated by DUX4-induced MBD3L proteins”. Since these are now gene names, we assume that they will not be considered abbreviations or acronyms. We would be glad to further modify the title if the reviewers or editor have other suggestions. We have also revised the Impact Statement and Abstract to include a description of the purification method.

11) Similarly, the schematic describing the purification could be included in Figure 1 proper, rather than in the supplemental data for Figure 1.

We have included the enChIP purification schematic in Figure 1 proper as Figure 1 and updated the Results and figure legends accordingly (see subsection “enChIP-MS identifies NuRD complex components as D4Z4 repeat-associated proteins”).

In addition to the changes noted above, we also deleted a paragraph in the Discussion of the original submission because we felt it did not add a sufficiently focused perspective to the work.